# Loss of POMC-mediated antinociception contributes to painful diabetic neuropathy

Divija Deshpande[1,2,3], Nitin Agarwal[2,11], Thomas Fleming[1,4,11], Claire Gaveriaux-Ruff [5,6,7,8,9], Christoph S. N. Klose [3], Anke Tappe-Theodor [2], Rohini Kuner [2] & Peter Nawroth[1,4,10 ✉]

Painful neuropathy is a frequent complication in diabetes. Proopiomelanocortin (POMC) is an endogenous opioid precursor peptide, which plays a protective role against pain. Here, we report dysfunctional POMC-mediated antinociception in sensory neurons in diabetes. In streptozotocin-induced diabetic mice the *Pomc* promoter is repressed due to increased binding of NF-kB p50 subunit, leading to a loss in basal POMC level in peripheral nerves. Decreased POMC levels are also observed in peripheral nervous system tissue from diabetic patients. The antinociceptive pathway mediated by POMC is further impaired due to lysosomal degradation of μ-opioid receptor (MOR). Importantly, the neuropathic phenotype of the diabetic mice is rescued upon viral overexpression of POMC and MOR in the sensory ganglia. This study identifies an antinociceptive mechanism in the sensory ganglia that paves a way for a potential therapy for diabetic neuropathic pain.

[1] Department of Medicine I and Clinical Chemistry, University Hospital of Heidelberg, INF 410 Heidelberg, Germany. [2] Institute of Pharmacology, Heidelberg University, INF 366, Heidelberg 69120, Germany. [3] Department of Microbiology, Infectious Diseases and Immunology, Charité -Universitätsmedizin Berlin, Hindenburgdamm 30, 12203 Berlin, Germany. [4] German Center for Diabetes Research (DZD), Neuherberg, Germany. [5] Institut de Génétique et de Biologie Moléculaire et Cellulaire, Department of Translational Medicine and Neurogenetics, Illkirch, France. [6] Université de Strasbourg, Illkirch, France. [7] Centre National de la Recherche Scientifique, UMR7104 Illkirch, France. [8] Institut National de la Santé et de la Recherche Médicale, U1258 Illkirch, France. [9] Ecole Supérieure de Biotechnologie de Strasbourg, Illkirch, France. [10] Joint Heidelberg-IDC Translational Diabetes Program, Helmholtz Zentrum, 85764 Neuherberg, Germany. [11] These authors contributed equally: Nitin Agarwal, Thomas Fleming. ✉email: peter.nawroth@med.uni-heidelberg.de

During diabetes the structure and function of nerves in the periphery are frequently perturbed, leading to sensorimotor neuropathy[1,2]. Approximately half of the patients with diabetic peripheral neuropathy (DPN) present chronic symptoms of neuropathic pain, including excruciating stabbing/burning sensation and allodynia which further cause anxiety, sleep disturbances, and lead to a poor quality of life[3–6]. Currently, the main lines of approach for treating painful DPN comprise drugs affecting synaptic noradrenaline and serotonin reuptake, sodium channel blockers, and opioids, apart from tight blood glucose control[7]. However, neither of these approaches can restore the sensory deficits of painful DPN to normality[8–11]. An improved analgesic treatment that provides pain relief by targeting specific pathways of neuropathic pain in DPN is therefore desirable over the currently available generalized therapeutic options. In order to develop such a treatment strategy, there is an urgent need to understand the less known patho-mechanisms underlying painful DPN.

The endogenous opioid system is our body's own line of defense against noxious stimuli. Endogenous opioid precursors (classically, proopiomelanocortin (POMC), pro-dynorphin (PDYN), and pro-enkephalin (PENK)) are proteolytically cleaved to produce opioid peptides. These bind to their cognate opioid receptors and trigger downstream signaling events (activation of $K^+$ channels and inhibition of $Ca^{++}$ channels), resulting ultimately in the inhibition of neuronal excitability. Persistent neuropathic pain in DPN patients suggests a dysfunction in endogenous opioid-mediated antinociceptive mechanisms. Previous studies have examined the link between the endogenous opioid system and diabetes, and reported that opioid peptide levels are altered in the central nervous system and plasma of diabetic rodent models[12–14] as well as of diabetic patients[15,16]. To date, however, it remains unknown whether the opioid levels in the peripheral nervous system (PNS) are altered during diabetes.

In the PNS, it is known that opioid receptors, including μ-opioid receptor (MOR), are expressed by neurons of the dorsal root ganglia (DRG) and carried to the nerve endings by anterograde axonal transport[17]. For determining the local source of opioids, studies have focused on animal models of neuropathic pain resulting from physical nerve trauma, which showed that immune cells infiltrating the injury site express opioid peptides[18–20]. However, the source of opioids in neuropathic pain caused by other triggers, such as damage induced by metabolic dysfunction in diabetes, remains uncharacterized.

Here, we report that the sensory neurons of the PNS express POMC in mice and humans and further demonstrate that the endogenous opioid pathway mediated by POMC is impaired during experimental and clinical diabetes. In vivo correction of this deficit is able to rescue diabetes-induced hyperalgesia and associated behavioral changes, thereby showing the relevance of the dysfunctional POMC-MOR signaling to the increased pain sensitivity observed in diabetes. This study, therefore, identifies a pivotal antinociceptive mechanism in the PNS and provides a therapeutic target for the treatment of painful DPN and possibly other painful peripheral neuropathies.

## Results

### POMC is expressed in the adult peripheral sensory neurons of mice and humans.
In order to understand the potential role of POMC-mediated antinociceptive signaling in the periphery, it was necessary to examine whether POMC is expressed in the PNS under basal conditions. We, therefore, analyzed different tissues of the PNS, namely lumbar DRG, sciatic nerves, and footpads of healthy adult C57Bl/6 mice for *Pomc* gene expression using primers detecting full-length *Pomc* mRNA (Supplementary Table T1). *Pomc* gene expression was predominantly observed in the lumbar DRG, while the gene expression in sciatic nerves and footpads was negligible in healthy mice (Fig. 1a).

To identify the cell type(s) expressing POMC peptide in the PNS, we co-stained lumbar DRG and sciatic nerves of healthy mice with anti-POMC antibody and cell-specific markers (anti-ß-tubulin III/Tuj-1 as pan-neuronal marker, anti-CD11b as pan-glial cell marker, and anti-CD45 as pan-immune cell marker) (Figs. S1 and S2). Confocal dual-immunofluorescence analysis revealed that the POMC-specific immunoreactivity was primarily localized in ß-tubulin III+ neuronal somata in the lumbar DRG and axons in sciatic nerves (Fig. 1b), supporting the concept that POMC is expressed in the peripheral sensory neuronal somata and then undergoes axonal transport. Among the different nerve populations within mouse DRG, ~60% of peptidergic putative nociceptors (CGRP +), ~30% non-peptidergic putative nociceptors (IsolectinB4 +), and ~55% of the large diameter myelinated neurons (NF200 +) expressed POMC (Fig. 1c, d). Importantly, POMC was also expressed in human lumbar DRG (Fig. 1c). Human DRG co-stained with POMC-specific antibody and the above-mentioned neuronal subtype markers demonstrated that the expression profile was consistent with the observations in mouse DRG (Fig. 1c, d, and Supplementary Table T2).

Specificity of the anti-POMC antibodies was confirmed using mouse brain (known to express POMC in hypothalamus) as positive control (Fig. S3a, b) and DRG from *Pomc* conditional knockout mice (cKO, with a neuron-specific *Pomc* gene deletion generated using Cre-Lox system) as the negative control (Fig. S3c, d). The loss of more than 95% of the POMC protein in the DRG of the cKO mice compared to the wild-type DRG detected by immunoblotting (Fig. S3d) further confirmed our observation that the DRG neurons are the primary source of POMC in the PNS.

Finally, to determine the association of POMC with pain signaling, we tested whether POMC responds to capsaicin-induced acute pain in the hindpaws of healthy mice (Fig. S4a). We observed that the unilateral capsaicin injection in the hindpaw acutely evoked nocifensive behavior (Fig. S4b). This stimulus resulted in an initial lowering of POMC protein level in the PNS indicating release of the neuropeptide (Fig. S4d), followed by normalization of the protein level (Fig. S4e) owing to the increased gene expression (Fig. S4c). This provided a functional link of POMC with the antinociceptive pathways in the PNS.

### POMC is downregulated in the PNS of diabetic mice.
We next addressed the regulation of POMC in diabetes by comparing its expression levels in streptozotocin (STZ)-induced diabetic mice and age-matched control mice (STZ-untreated). We employed the low dose STZ model, which is not associated with overt acute toxicity[21]. In comparison with the control mice, STZ-induced mice showed increased blood glucose levels, HbA1c content, and decreased body weight (Supplementary Tables T3 and T4), parameters characteristic of diabetes. A time-course study of the diabetic mice revealed that the female mice developed significant heat hyperalgesia at 12 weeks (Fig. S5a) and the male mice at 6 weeks post STZ-treatment (Fig. S5b). Respective optimal time-points were studied further for either gender. We employed female mice to elucidate the molecular mechanisms underlying increased pain sensitivity. The main findings observed in the female mice were further verified in the male mice.

The diabetic mice displayed significantly increased sensitivity not only to heat (Figs. 2b and S6b), but also to mechanical stimuli (Figs. 2a and S6a). Interestingly, we found that at the time-point

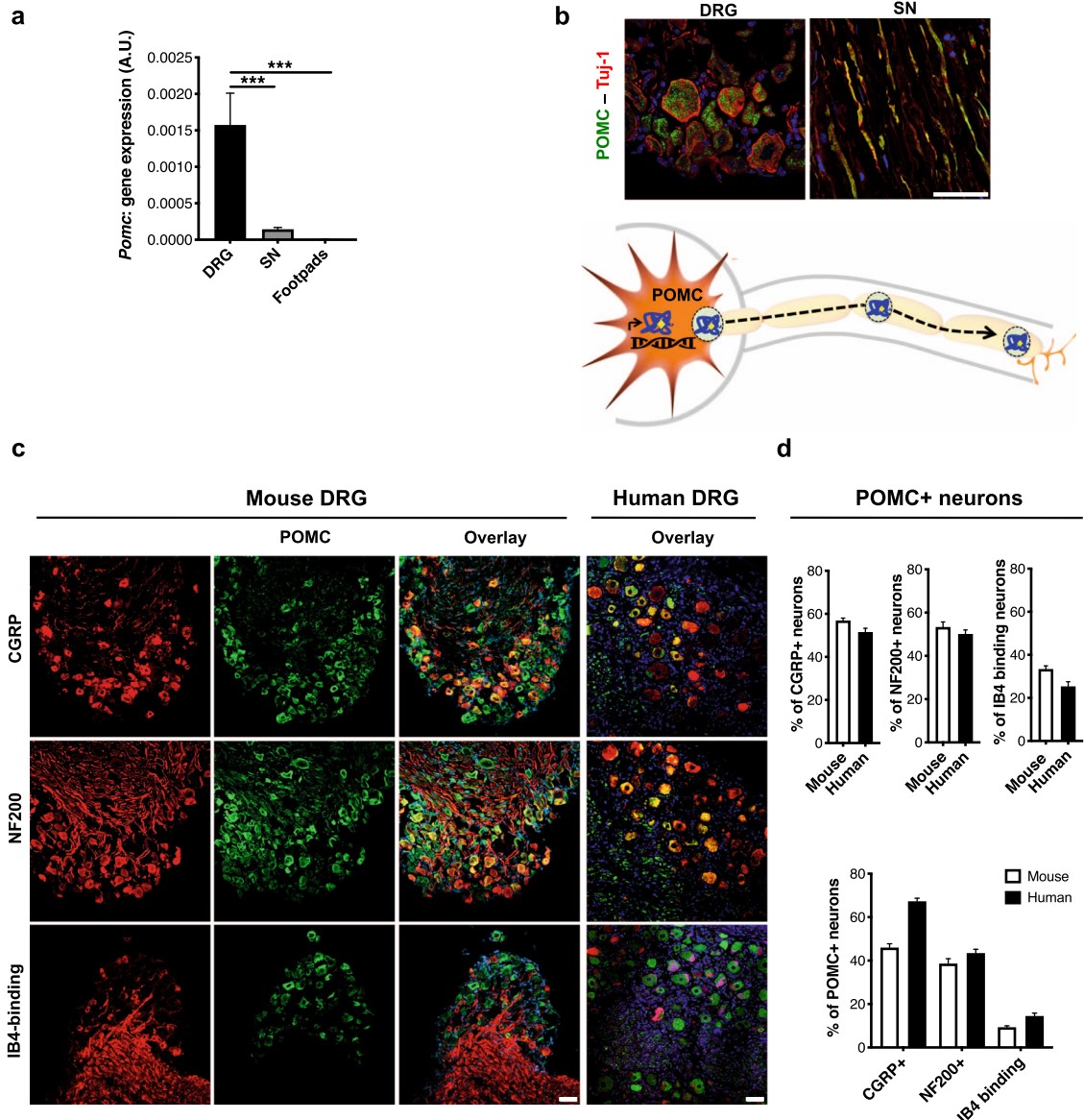

**Fig. 1 POMC is expressed by the peripheral sensory neurons in healthy mice and humans. a** *Pomc* gene expression (18srRNA normalized) quantified in lumbar DRG, sciatic nerves (SN) and footpads of healthy C57Bl/6 mice (*n* = 5 mice, 2 replicates from each mouse; one-way ANOVA followed by Tukey's post-hoc test; ***$p$ < 0.0001) **b** Expression of POMC protein (Rb anti-POMC antibody) and its colocalization with Tuj-1+ neurons in mouse lumbar DRG and SN (*n* = 6 mice, 4–6 sections per tissue from each mouse) and, schematic showing expression and axonal transport of POMC in the PNS. **c** Typical examples and **d** quantitative analysis of the distribution of neuronal-subtypes expressing POMC (Gt anti-POMC antibody) via co-immunolabeling with neuronal markers protein in lumbar DRG of healthy mice (*n* = 6, 4–10 DRG sections from each mouse) and healthy humans (*n* = 6 subjects, 4–10 DRG sections from each human subject). For all panels, data represent mean ± SEM; with 95% C.I. Circles represent individual data points. Scale = 50 μm. Source data are provided as a Source Data file.

of heightened pain sensitivity, *Pomc* gene expression was downregulated in the DRG of diabetic mice (Fig. 2c). In the control mice, anti-POMC immunoreactivity was primarily localized in the neuronal somata, and in the axons (Figs. S1a and S2a) in diabetic mice. This finding established that neurons are the major cell type to express POMC in the PNS under healthy and diabetic conditions (Figs. S1 and S2). However, the number of neurons expressing POMC, as well as anti-POMC signal intensity measured using ImageJ tools, was significantly lower in the DRG of diabetic mice (Figs. 2d and S6a, b). We further quantified these differences by immunoblotting and observed a decrease in the density of the ~26 KDa band corresponding to POMC peptide in the total lysates of lumbar DRG, sciatic nerves, and footpads of diabetic mice (Figs. 2e,

S6c–e, and S7c, d). Additionally, we also found that the basal level of the ß-endorphin (opioid peptide cleaved proteolytically from POMC) had also declined in the PNS of diabetic mice, notably in the footpads, wherein maximum content of the opioid peptide was detected in the control mice (Fig. 2f). Importantly, these changes did not result from reduced levels or integrity of the diabetic tissue since we had normalized for expression of control proteins. These data indicate that POMC levels drop throughout the PNS in diabetes.

**MOR is downregulated in the PNS of diabetic mice**. Functionality of the POMC-mediated antinociceptive pathway depends upon the ligand (ß-endorphin), as well as its receptor (MOR).

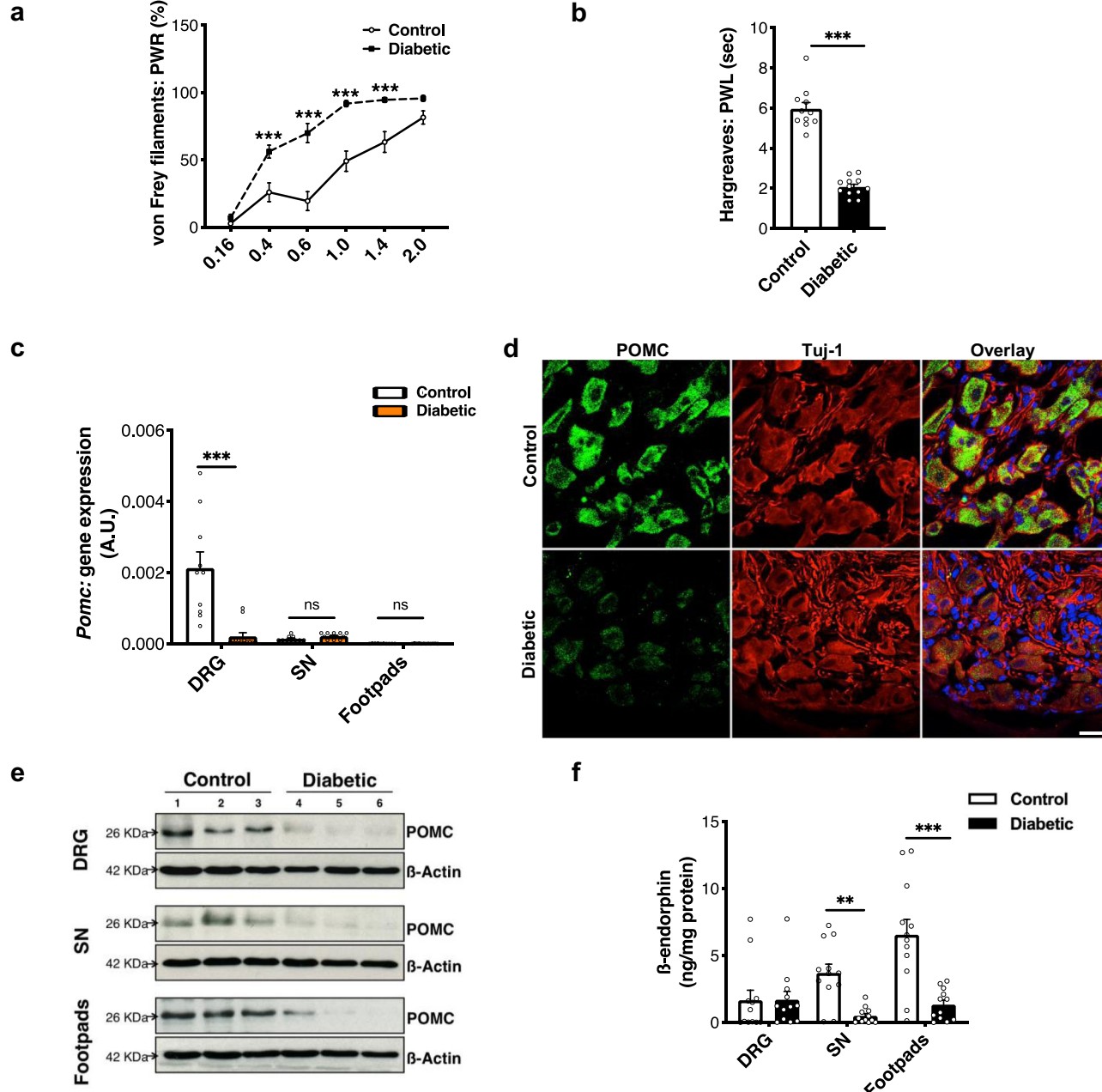

**Fig. 2 POMC is downregulated in the PNS of mice with experimental diabetes.** Measuring 12 weeks STZ-treated diabetic mice and age-matched controls (STZ-untreated) for: **a** mechanical hyperalgesia (control, $n = 13$ and diabetic, $n = 18$; two-way ANOVA followed by Sidak's post-hoc test; ***$p < 0.001$). PWR = paw withdrawal responses and **b** thermal hyperalgesia (control, $n = 11$ and diabetic, $n = 12$; two-tailed $t$-test; ***$p < 0.0001$). PWL = paw withdrawal latency (circles). **c** *Pomc* gene expression (18srRNA normalized) quantified in lumbar DRG, sciatic nerves (SN), and footpads ($n = 5$–6 mice/group; two-way ANOVA followed by Sidak's post-hoc test; ***$p < 0.0001$). **d** Typical examples of co-immunostainings showing POMC protein expression (Rb anti-POMC antibody) in Tuj-1+ neurons in the DRG of control and diabetic mice ($n = 5$ mice/group, 5–9 DRG sections from each mouse) **e** Representative blots showing comparison of POMC protein level (using Gt antibody) in total lysates of lumbar DRG, SN, and footpads (control, $n = 9$ and diabetic, $n = 9$) **f** ß-endorphin level in lumbar DRG, SN, and footpads of control and diabetic mice quantified by ELISA ($n = 6$ mice/group, 2 replicates from each mouse; two-way ANOVA followed by Sidak's post-hoc test; **$p = 0.0049$, ***$p < 0,0001$). For all panels, female mice data are shown; data represent mean ± SEM; with 95% C.I. Circles represent mean value per mouse in panel (**b**) and individual data points in panels (**c**, **f**). For panel (**a**), solid line with circles represents control mice and dotted line with black squares represents diabetic mice. For panel (**c**), white bar: control and orange bar: diabetic mice. Scale = 20 μm. Source data are provided as a Source Data file.

In the PNS, MOR is known to be expressed by sensory neurons in the DRG[18,22]. Therefore, we quantified mRNA and protein levels of MOR in the lumbar DRG. No significant difference was observed between control and diabetic mice with respect to mRNA levels (Fig. 3a). Surprisingly, the MOR protein level, as detected by immunoblotting, was significantly lower in the DRG of diabetic mice (Figs. 3b and S7e, f). Supporting this, we found significantly fewer ß-tubulin III+ neurons expressed MOR in the DRG of diabetic mice (Fig. 3c). Signal obtained by the anti-MOR antibody in immunostaining and immunoblotting is shown to be specific using MOR$^{-/-}$ DRG (Fig. S8a, b). The specificity of this antibody has also been demonstrated previously[22].

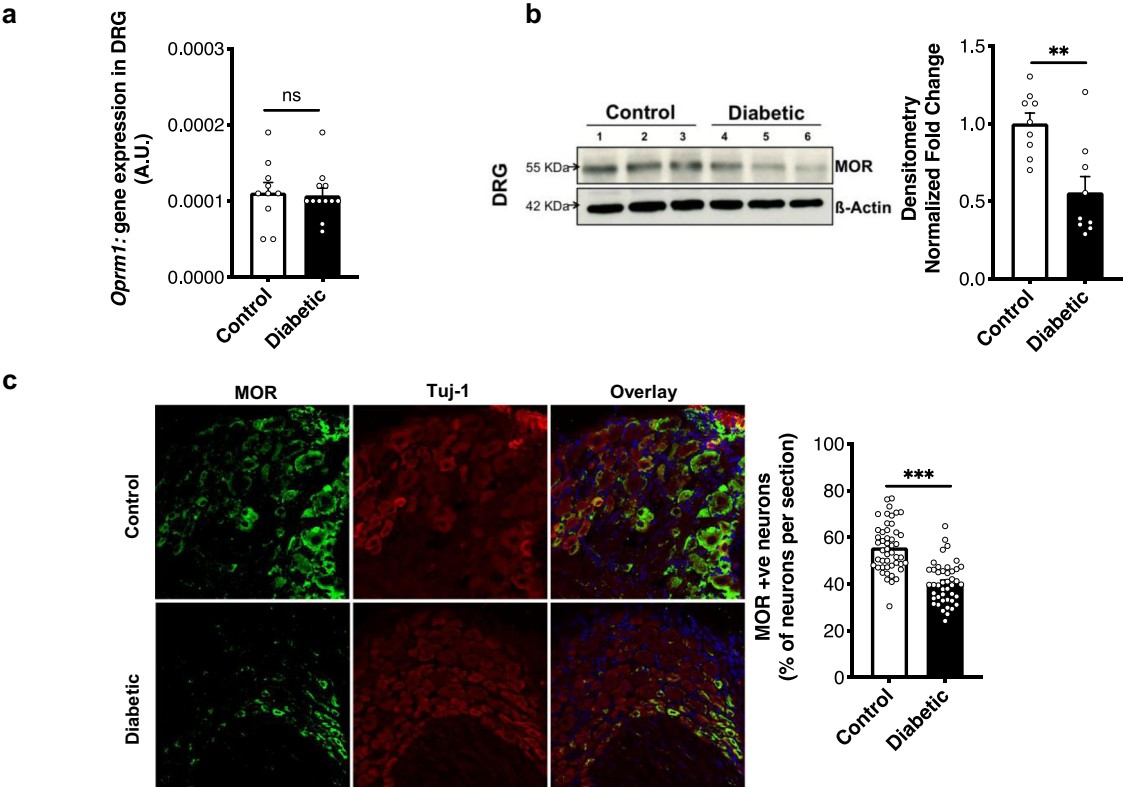

**Fig. 3 MOR is downregulated in the PNS of mice with experimental diabetes. a** *Oprm1* gene expression (18srRNA normalized) compared in lumbar DRG of control and 12 weeks diabetic mice (*n* = 5–6 mice/group, 2 replicates from each mouse; two-tailed *t*-test; n.s. = non-significant, *p* = 0.4) **b** Representative blot and densitometric quantification of MOR protein (∼55 KDa band normalized to actin) in total lysates of lumbar DRG (*n* = 9 mice/ group; two-tailed *t*-test; \*\**p* = 0.002). **c** Typical examples and quantitative analysis showing MOR protein expression in Tuj-1+ neuronal cells in the lumbar DRG of control and diabetic mice (*n* = 6 mice/group; 8–10 DRG sections from each mouse; two-tailed *t*-test; \*\*\**p* < 0.0001). For all panels, female mice data are shown; data represent mean ± SEM; with 95% C.I. Circles represent individual data points. Scale = 20 μm. Source data are provided as a Source Data file.

Thus, these observations suggest that the POMC-MOR axis is depressed at both ligand and receptor levels in the PNS of diabetic mice at a time-point corresponding to nociceptive hypersensitivity.

**High glucose represses *Pomc* promoter.** To understand the molecular mechanism(s) underlying reduced *Pomc* transcription in diabetes, we performed in vitro promoter studies in AtT-20 cells (known for constitutive POMC expression[23]). In silico analysis showed that the *Pomc* promoter has the NF-kB binding region in close proximity to the corticotrophin-releasing hormone (CRH) binding region (Fig. 4a). CRH is a known *Pomc* promoter agonist, while the role of NF-kB in regulating *Pomc* promoter remains unclear[23]. We therefore addressed whether NF-kB might influence *Pomc* promoter activity under high glucose conditions.

To test this concept, we exposed AtT-20 cells to increasing glucose concentrations (5, 10, and 20 mM) for 12 h and noted a corresponding increase in the nuclear localization and DNA binding ability of NF-kB (Fig. 4c), suggesting increased NF-kB activation. Using supershift EMSA, p50 subunit of NF-kB was identified to be activated under high glucose condition (Fig. S9). This finding was supported by increased accumulation of p50 antigen in the nuclear lysates of cells exposed to high glucose (20 mM) for 12 h, as compared to those exposed to normal glucose (5 mM) (Fig. 4c). ChIP analysis of the cells exposed to high glucose for 12 h confirmed an increased binding of p50 NF-kB subunit to the *Pomc* promoter, as compared to cells exposed to normal glucose (Fig. 4d). The effect of this increased

p50–promoter binding on the promoter activity was examined by dual-luciferase reporter assay. The cells were transfected with either wild-type (WT) *Pomc* promoter or *Pomc* promoter mutated at NF-kB binding site (Mutant), and were then exposed to normal (5 mM) and high glucose (20 mM) for 12 h. The promoter activity (WT and Mutant) escalated in the presence of CRH under normal glucose condition. However, under high glucose condition, the response of the WT promoter to CRH was diminished. In contrast, the Mutant promoter retained its responsivity to CRH even under high glucose conditions (Fig. 4e).

Similar to the in vitro findings, a 2-fold increase in the binding of NF-kB p50 to *Pomc* promoter was detected in the lumbar DRG of diabetic mice (Fig. 4f). A co-staining with anti-p50 and anti-POMC antibodies in the DRG of diabetic mice showed that POMC immunoreactivity was decreased in those neurons which showed an increased p50 signal, supporting a negative relationship between NF-kB p50 protein and POMC expression under hyperglycemic conditions (Fig. 4g).

**PKC activation in diabetes causes MOR degradation.** As the expression of *Oprm1* gene (MOR protein-coding gene) remained unchanged (Fig. 3a) in the lumbar DRG during diabetes, we hypothesized that the loss of MOR was due to its degradation. Phosphorylation of MOR precedes its internalization from the cell surface and its subsequent lysosomal targeting[17]. MOR can be phosphorylated by kinases, such as, G-protein-related kinases in the presence of a selective MOR agonist[17] or by protein kinase C (PKC), in the absence of an agonist[24]. Moreover, activation of

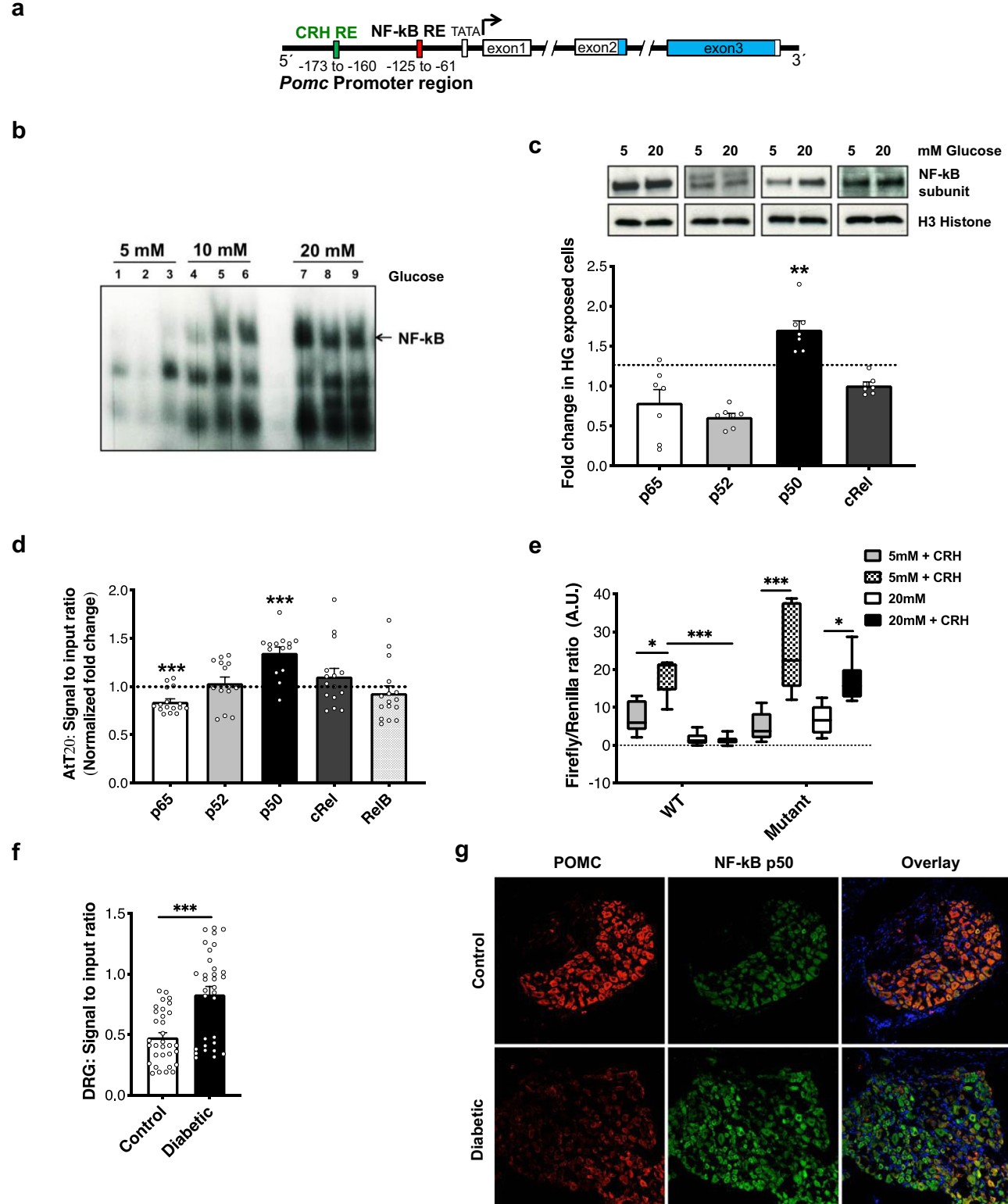

PKC is regarded as a hallmark of late stages of diabetes[25]. Therefore, to address whether PKC might play a role in MOR degradation during diabetes, we quantified PKC activation in the DRG. A significantly increased PKC activation was noted in the DRG of diabetic mice (Fig. 5a). Furthermore, it is reported that PKC phosphorylates MOR at Thr370 residue[24]. We, therefore, examined MOR phosphorylation by immunoprecipitating MOR protein from the lumbar DRG and using a pan-phosphothreonine

antibody for detection by immunoblotting. Indeed, an increased level of threonine-phosphorylated MOR was detected in the DRG of diabetic mice (Fig. 5b). The antibody used for immunoprecipitating MOR has been demonstrated to be specific previously[26,27].

Subsequently, we questioned whether a sustained PKC activation, which occurs in a hyperglycemic environment, could affect the internalization of MOR. For this purpose, studies were

**Fig. 4 High glucose represses *Pomc* promoter. a** Schematic representation of *Pomc* promoter region containing the NF-kB binding region in close proximity to the CRH binding region. **b** NF-kB activation measured as a function of its DNA binding ability, using EMSA, in the nuclear lysates of AtT-20 cells exposed to increasing glucose concentrations for 12 h. **c** Representative blots and densitometric quantification of NF-kB subunit proteins in the nuclear lysates of AtT-20 cells exposed to normal (5 mM; NG) or high (20 mM; HG) glucose conditions for 12 h, using subunit-specific antibodies. Bands detected at p65 (~65 KDa), p52 (~60 KDa), p50 (~50 KDa), cRel (~72 KDa). Histone H3 (~10 KDa) normalization used to obtain densitometric ratio for each subunit ($n$ = total 7 replicates/group taken from two independent experiments; two-tailed $t$-test for comparison between NG and HG exposed cells per subunit; **$p$ = 0.002). Dotted line represents basal level for each subunit in cells exposed to NG. **d** Binding of each NF-kB subunit to *Pomc* promoter in AtT-20 cells exposed to NG or HG for 12 h ($n$ = total 14–16 replicates/group taken from 4 independent ChIP assays; two-tailed $t$-test for comparison between NG and HG exposed cells per subunit;***$p$ < 0.0001). Dotted line represents binding of each NF-kB subunit in cells exposed to NG. **e** *Pomc* promoter activity measured in AtT20 cells transfected with a wild-type *Pomc* promoter-luciferase construct (WT) or a promoter construct mutated at NF-kB binding site (Mutant). Promoter activity quantified under NG or HG exposure condition (12 h) as function of response to the promoter agonist, corticotropin-releasing hormone (CRH; $10^{-8}$ M, 6 h) ($n$ = total 7 replicates per exposure condition taken from 2 independent experiments; two-way ANOVA followed by Sidak's post-hoc test; *$p$ = 0.01, ***$p$ < 0.0001). **f** ChIP assay in lumbar DRG of control and diabetic mice to quantify the binding of NF-kB p50 to *Pomc* promoter in vivo ($n$ = 8 female mice/group; two-tailed $t$-test;***$p$ < 0.0001). **g** Typical examples of co-immunostaining using antibodies for NF-kB p50 and POMC (Gt antibody) in the lumbar DRG of control and diabetic mice ($n$ = 6 female mice/group, 3–7 DRG sections from each mouse). For panels (**c**), (**d**), (**f**), data represent mean ± SEM; circles represent individual data points. The boxplot in panel (**e**) corresponds to median, 25th, and 75th percentiles, whiskers correspond to minimum and maximum values. Scale = 50 μm. For all panels, 95% C.I. Source data are provided as a Source Data file.

conducted using in vitro DRG cultures. Mouse DRG neurons are normally cultivated at glucose conditions of 10–17 mM glucose in most studies[28–30]. Therefore, we exposed the cells to 40 mM for 48 h to simulate hyperglycemia and examined the activation of PKC, determined by its subcellular localization[31]. In an active state, PKC immunoreactivity was associated with the cell surface denoted by wheat germ agglutinin (WGA). Under high glucose conditions, the number of neuronal cells showing association of PKC with WGA was significantly higher (Fig S10a).

To study subcellular localization of MOR during PKC activation, primary DRG cells transfected with MOR-mCherry were exposed to normal (NG, 17 mM) or high glucose (HG, 40 mM) conditions in the presence or absence of PKC inhibitor (Gö6983, 1 μM) for 48 h. MOR-mCherry signal was associated with WGA-stained cell surface under NG condition to a similar extent in the absence or presence of the PKC inhibitor. However, when the cells were exposed to HG, MOR-mCherry signal was found to localize within the cells rather than being associated with cell membrane. This internalization of MOR was significantly reduced in the presence of the PKC inhibitor (Fig. 5c).

Finally, to directly address whether MOR was lysosomally degraded during diabetes, DRG were co-stained with anti-MOR antibody and lysosomal marker, Lamp-1. anti-MOR immunor-eactivity was localized within the neuronal cell bodies and associated Lamp-1 to a greater extent in DRG of diabetic mice compared to controls (Fig. 5d). Thus, PKC activation appears to be associated with MOR degradation under in vitro and in vivo hyperglycemic conditions.

**Overexpression of POMC and MOR reverses mechanical hypersensitivity and its aversive effect in diabetic mice.** Having observed a diabetes-associated downregulation of the endogenous POMC-MOR, it was necessary to establish whether restoration of this axis could be a therapeutic strategy for DPN. Therefore, we overexpressed both proteins, POMC and MOR, either singly or in combination in the DRG of the diabetic mice and tested them for mechanical sensitivity. Specifically, the right (ipsilateral) L3 and L4 DRG of diabetic mice were injected directly with rAAV-1/2 constructs expressing MOR-GFP, POMC-GFP, or bicistronic POMC-MOR-FLAG (Fig. 6a). Mice with unilateral DRG injections of only GFP-expressing viral constructs were included as an internal control group to rule out any potential impact of DRG injections or viral expression itself. The left (contralateral) L3 and L4 of all mice did not receive virus injections. The AAV-injected mice were then treated with STZ and tested for behavior. The

same mice were also tested for behavior 45 min after intraplantar injection of PKC inhibitor (Gö6983, 20 μM).

We immunostained the AAV-injected DRG post-mortem with anti-GFP or anti-FLAG antibodies and found that 41 ± 8.1% of neurons per DRG section were infected with virus particles expressing either of the constructs (Fig. 6b). This is consistent with our previous reports showing persistent infection of both the large and small diameter neurons of the lumbar DRG with rAAV 1/2 serotype particles[32].

In mechanical sensitivity test using Frey filaments, contralateral hindpaw of all diabetic mice displayed heightened pain sensitivity compared to the naïve controls (i.e., non-diabetic and without any AAV injection) at all forces applied (Fig. 6c and Supplementary Table T5). PKC inhibitor did not alter the mechanical sensitivity of the contralateral hindpaw in any mice cohorts (Fig. 6e and Supplementary Table T7). Observations were similar with respect to the ipsilateral hindpaw of mice overexpressing either GFP or MOR (Fig. 6d, f and Supplementary Tables T6 and T8). However, the ipsilateral hindpaw of mice overexpressing POMC or those overexpressing POMC-MOR combination, demonstrated ameliora-tion in mechanical hypersensitivity upto 1.4 g force (Fig. 6d and Supplementary Table T6) suggesting a modulation of diabetes-associated mechanical allodynia in female (Fig. 6) and male (Fig. S11) mice. There was no significant difference in the responses between the cohorts overexpressing POMC alone and POMC-MOR combination in the ipsilateral hindpaw at any force. PKC inhibition did not augment the hypo-nociceptive phenotype seen in the POMC-MOR expressing mice compared to the mice expressing POMC alone (Fig. 6f and Supplementary Table T8).

We also analyzed the changes in aversion-related behavior of the diabetic mice in response to mechanical stimulation and obtained consistent results. We performed place escape/avoidance paradigm (PEAP), in which separate hindpaws were continuously stimulated depending on whether the mouse was in the light or dark chamber (Fig. 6g), and time spent in the light chamber was recorded. In this test, the naïve control mice did not react to the mechanical stimulation and preferred the dark chamber as time passed. The diabetic mice expressing GFP alone, responded to the mechanical stimulus on both paws and spent increasingly more time in the dark chamber, demonstrating that these mice preferred the dark over light chamber if both were equally aversive. However, the diabetic mice expressing POMC or POMC-MOR combination in their right hindpaw (ipsilateral) spent significantly more time in the light chamber compared to the diabetic mice expressing only GFP (Fig. 6h and Supplementary Table T9). Thus, only the diabetic mice with POMC or POMC-MOR overexpression in their right hindpaw

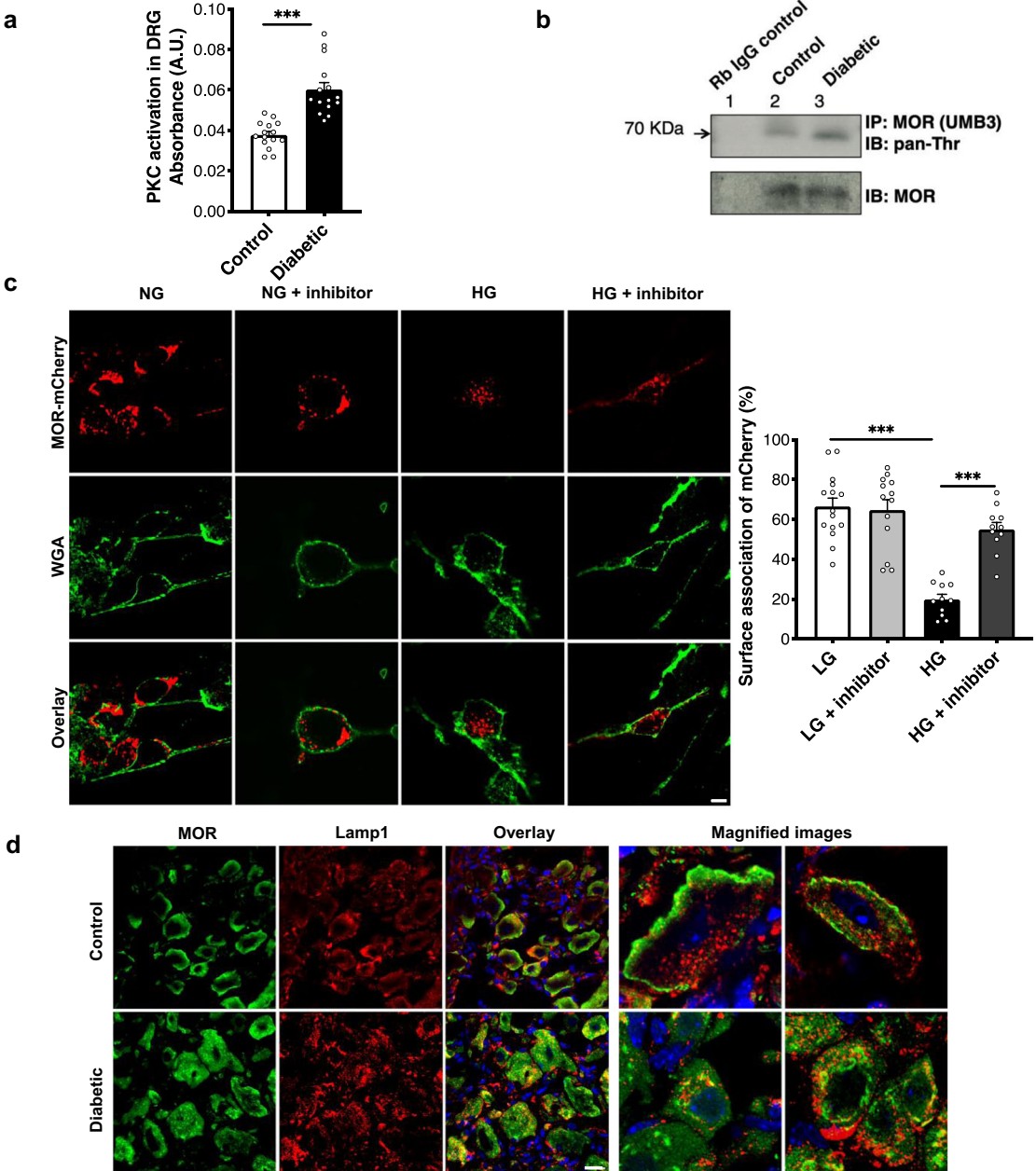

**Fig. 5 PKC activation during diabetes causes MOR degradation in lumbar DRG. a** Kinase activity assay to compare PKC activation in the lumbar DRG of control and diabetic mice ($n = 8$ mice/group, 1–2 replicates from each mouse; two-tailed $t$-test; ***$p < 0.0001$). **b** Lumbar DRG protein lysates from control and diabetic mice ($n = 3$ each) pooled and immunoprecipitated(IP) using MOR-specific antibody (UMB3) and immunoblotted(IB) using pan-phosphothreonine antibody to determine phosphorylation of MOR. **c** In vitro, primary DRG neurons transfected with MOR-mCherry construct and exposed to normal glucose (NG, 17 mM) or HG (high glucose, 40 mM) in the presence or absence of a PKC inhibitor (Gö6983, 1 μM, 48 h). Surface association of MOR assessed by overlap of WGA-stained cell membrane and mCherry signal ($n =$ total of 11–15 images analyzed from 3 independent experiments; one-way ANOVA followed by Tukey's post-hoc test; ***$p < 0.0001$). **d** Typical examples showing co-immunostaining of control and diabetic lumbar DRG sections with a MOR-specific antibody and lysosomal marker (Lamp1) ($n = 6$ mice/group, 3–6 DRG sections from each mouse). For all panels, female mice data are shown; data represent mean ± SEM; with 95% C.I. Scale = 20 μm. Circles represent individual data points. Source data are provided as a Source Data file.

made a 'conscious' decision to escape a preferred dark environment to avoid mechanical pain in their contralateral hindpaw. The light–dark preference of all mice remained the same post PKC inhibitor injections (Fig. 6i and Supplementary Table T10) suggesting a limited role of PKC-controlled MOR in diabetic mechanical allodynia and its aversion.

In summary, overexpression of either POMC or the combination of POMC-MOR alleviated neuropathic mechanical hyperalgesia and its aversion in diabetic mice to a similar degree.

**Genetic reconstitution of POMC and MOR rescue thermal hyperalgesia in diabetic mice.** In the Hargreaves assay (Fig. 7a), naïve control mice showed no difference in thermal sensitivity between both hindpaws. Contrastingly, both hindpaws of diabetic mice overexpressing only GFP showed a significant thermal hyperalgesia compared to naïve controls. Mice overexpressing MOR responded in a similar manner as the mice overexpressing only GFP. The pain responses of both hindpaws of naïve controls and diabetic mice overexpressing GFP or MOR remained

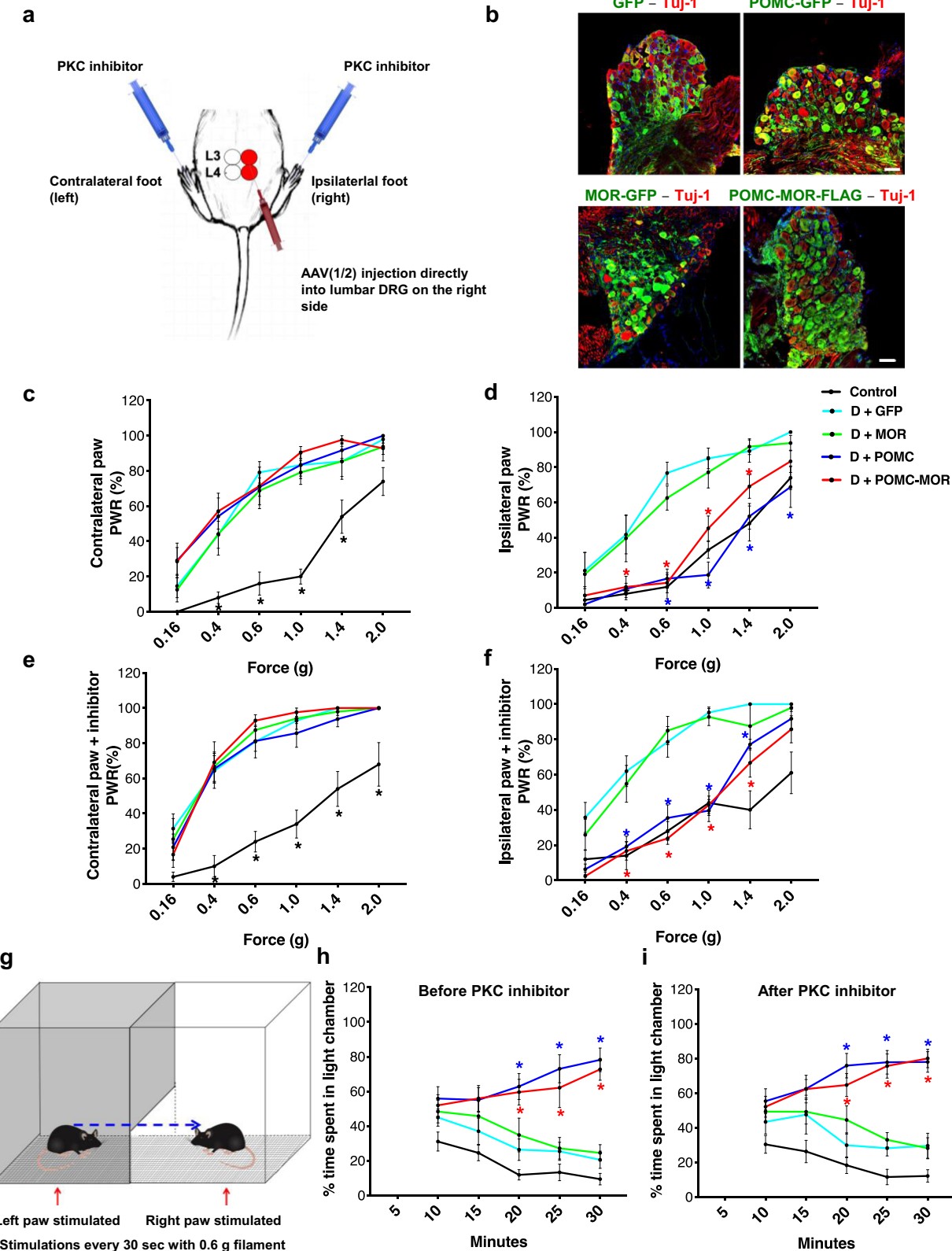

unchanged when measured after intraplantar injection of the PKC inhibitor.

However, in the ipsilateral hindpaw of mice overexpressing either POMC or the combination of POMC-MOR in DRG neurons, there was a significant reduction of approximately 3-fold in thermal hyperalgesia compared to their respective contralateral hindpaw. Remarkably, this response of the

ipsilateral hindpaw of diabetic mice overexpressing both POMC and MOR improved by approximately 4-fold in the presence of PKC inhibitor. The protective effect imparted by the PKC inhibitor with respect to heat hyperalgesia was significantly greater in magnitude in the diabetic mice overexpressing POMC-MOR combination than in the diabetic mice over-expressing POMC alone.

**Fig. 6 Genetic reconstitution of POMC rescues mechanical hyperalgesia and its aversive effects. a** Schematic representation of direct DRG injection in diabetic mice with rAAV-1/2 constructs for overexpressing GFP, POMC, MOR, or POMC-MOR. Readings taken before and after intraplantar injection of PKC inhibitor in both hindpaws (Gö6983, 20 μM, 45 min). **b** Immunostaining of AAV-injected DRG with GFP- or FLAG-specific antibodies showing expression of the viral constructs (n = 8 mice/group). **c–f** Mechanical hypersensitivity measured using von Frey filaments in naïve control (non-diabetic, non-AAV injected) and AAV-1/2-injected diabetic mice. Pain measurements performed before (**c, d**) and after (**e, f**) intraplantar injection of PKC inhibitor (naïve control, n = 10; D + GFP, n = 7-8; D + MOR, n = 8; D + POMC, n = 8; D + POMC-MOR, n = 7). PWR = paw withdrawal responses. **g** Schematic representation of place escape/avoidance paradigm (PEAP) test. The % of time spent in the light chamber measured, **h** before and **i** after injection of PKC inhibitor in both hindpaws (naïve control, n = 11; D + GFP, n = 8; D + MOR, n = 10; D + POMC, n = 9–10; D + POMC-MOR n = 10). For panels (**c** and **e**), *black star indicates significant difference between readings of each mice cohort and readings of control mice. For panels (**d, f, h, i**), *blue star indicates significant difference between readings of D + POMC and D + GFP cohort, while *red star indicates significant difference between D + POMC-MOR mice and D + GFP mice readings. For all panels, female mice data are shown; data represent mean ± SEM; two-way ANOVA followed by Dunnett's post-hoc test; *p < 0.05 with 95% C.I. Cyan line: D + GFP, green line: D + MOR, blue line: D + POMC, red line: D + POMC-MOR. Scale=50 μm. p-values are given in Supplementary Tables T5–T10. Source data are provided as a Source Data file.

To further confirm this finding, we examined the diabetic mice expressing both POMC and MOR for thermal sensitivity using Hotplate test. Since unilaterally injected mice cannot be used for this test, we intrathecally injected rAAV-9/2 particles expressing either bicistronic POMC-MOR-FLAG construct or GFP construct to achieve expression in both hindlimbs (Fig. 7b, c). We observed a maximum reduction in thermal hyperalgesia in diabetic mice overexpressing POMC-MOR combination when injected with PKC inhibitor substantiating our findings in the Hargreaves test (Fig. 7d). Consistent results were obtained in the male diabetic mice (Fig. S12) as in the female diabetic mice (Fig. 7).

Thus, POMC-MOR combination coupled with PKC inhibition offered the highest antinociceptive effect against heat hyperalgesia.

**Genetic reconstitution of POMC and MOR is necessary to improve non-reflexive behavior in diabetic mice.** After observing the efficacy of restoring POMC and MOR against evoked hyperalgesia during diabetes, we further investigated their role in other pain-related non-reflexive behavior by performing gait analysis. The Catwalk system was used to analyze the gait parameters of the same unilaterally injected mice cohorts described above. Compared to the naïve controls, the diabetic mice showed significant decrease in parameters corresponding to paw pressure, namely, paw contact intensity, maximum contact area, and paw print area. Diabetic mice also demonstrated significantly increased stride length and swing indicative of slower gait compared to the naïve controls (Fig. 8a). These gait anomalies remained unchanged after receiving an intraplantar injection of PKC inhibitor (Gö6982, 20 μM) (Fig. 8c and Movies 1 and 2). However, an increase in paw contact intensity, paw print area, and maximum contact area was observed in the male (Fig. S13) and female (Fig. 8d and Movies 3 and 4) diabetic mice overexpressing POMC-MOR post PKC injection. This indicated that these mice put more pressure/limped on their less painful ipsilateral paw than their hypersensitive contralateral paw. A reduced swing was also noted in the same cohort.

Taken together, these findings show that restoring POMC and MOR in the peripheral sensory neurons emerge as a common effective therapeutic strategy against evoked (mechanical and heat) as well as spontaneous neuropathic hypersensitivity during diabetes.

**POMC and MOR are downregulated in the nerves of type-1 and type-2 diabetic patients.** Finally, we sought to validate our preclinical findings in diabetic patients. We immunostained the sciatic nerves of type-1 and type-2 diabetic patients with either anti-POMC or anti-MOR antibodies along and compared the signal intensities with a non-diabetic patient cohort of similar

demographics (Supplementary Table T11). Both POMC and MOR were remarkably decreased by a magnitude of 50% in ß-tubulin III+ nerves of diabetic patients, irrespective of the type of diabetes or gender (Fig. 9) compared to non-diabetic subjects. These changes did not result from reduced levels or integrity of the diabetic tissue, since we had normalized for expression of control proteins (Fig. S14). Immunoflourescence signal obtained from the anti-MOR antibody used here was demonstrated to be specific using MOR$^{-/-}$ DRG (Fig. S8c).

Thus, the confirmation of our preclinical findings into a clinical setting at a molecular level further strengthens the therapeutic significance of POMC-MOR axis in the PNS, specifically in the context of painful DPN.

## Discussion

Our study demonstrates the presence of the endogenous opioid precursor, POMC, in the sensory ganglia of mice and humans and its importance in antinociception, especially during painful DPN. We noted that during diabetes, the POMC-MOR opioid pathway was dysfunctional at the level of ligand as well as the receptor. This would provide an explanation why MOR-targeting exogenous opioids, such as tramadol and oxycodone, fail to produce analgesic relief uniformly in all DPN patients without causing side effects[33–35]. This study further points to the occurrence of molecular alterations occurring specifically in the sensory ganglia, which are responsible for downregulation of POMC and MOR in diabetes. Restoring the ligand–receptor pair in the lumbar DRG alleviated the neuropathic hypersensitivity and related behavior changes during experimental diabetes, pointing towards a therapeutic approach that targets the pathogenesis of painful DPN.

It is well-known that functional MOR is expressed by the neuronal somata in the sensory ganglia and is axonally transported towards the nerve endings under normal physiological conditions[18,36]. Yet, the common conception is that there are no local sources of opioids in the PNS under the same condition, despite the early reports on the presence of several opioid mRNAs and peptides (namely dynorphin, enkephaphalin and endomorphin) in the PNS[18]. Opioid presence in sensory neurons has largely been ignored; the primary reason being the lack of knowledge of their functional role in the PNS. To the best of our knowledge, this is the first study showing the expression and functional role for POMC in the PNS.

Our data show that the expression of POMC in the PNS, like MOR, is mainly somatic. Consequently, DRG soma is the main reservoir for the precursor POMC, whereas the likely site of endogenous MOR activation is nerve endings in the footpads, wherein maximum content of POMC-derived ß-endorphin, is detected. Such a concept of opioid precursor processing in the nerve endings is reinforced by several studies[37,38]. ß-endorphin

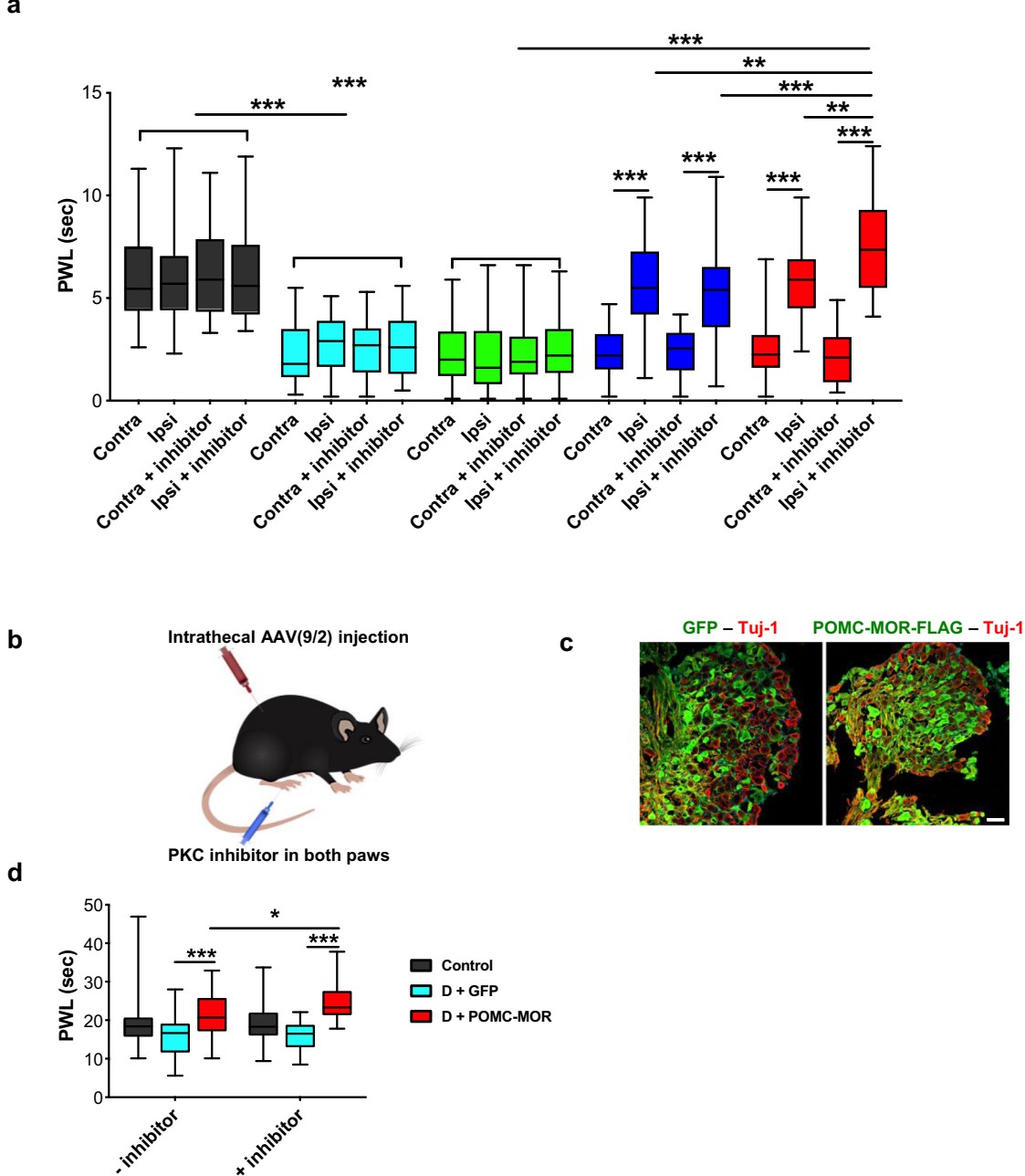

**Fig. 7 Genetic reconstitution of POMC and MOR rescues thermal hyperalgesia in diabetic mice. a** Thermal hypersensitivity measured using Hargreaves test in naïve controls (non-diabetic, non-AAV injected) and diabetic mice overexpressing rAAV-1/2 constructs only in their right (ipsilateral) L3 and L4 DRG. Readings taken before and after intraplantar injection of PKC inhibitor in both hindpaws (Gö6983, 20 μM, 45 min) (naïve control, $n = 10$; D + GFP, $n = 9$; D + MOR, $n = 10$; D + POMC, $n = 9$; D + POMC-MOR, $n = 10$). **b** Schematic representation of intrathecal injection in diabetic mice with rAAV-9/2 constructs for overexpressing GFP or POMC-MOR. Readings taken before and after intraplantar injection of PKC inhibitor in both hindpaws (Gö6983, 20 μM, 45 min). **c** Immunostaining of DRG with GFP- or FLAG-specific antibodies after intrathecal injection showing expression of the viral constructs ($n = 9–10$ mice/group). **d** Thermal hypersensitivity measured using Hotplate test in naïve controls and diabetic mice overexpressing rAAV-9/2 constructs (naïve control, $n = 10$; D + GFP, $n = 9$; D + POMC-MOR, $n = 9$). PWL = paw withdrawal latency. For all panels, female mice data are shown. The boxplot in panels (**a**) and (**d**) corresponds to median, 25th, and 75th percentiles, whiskers correspond to minimum and maximum values; two-way ANOVA followed by Tukey's post-hoc test; $*p = 0.040$, $**p < 0.01$, $***p < 0.0001$ with 95% C.I. Cyan bar: D + GFP, green bar: D + MOR, blue bar: D + POMC, red bar: D + POMC-MOR. Scale = 50 μm. Source data are provided as a Source Data file.

released at the nerve endings in the footpads may activate MOR in an autocrine and possibly in a paracrine fashion to inhibit the nociceptive signals generated at the periphery. The physiological outcome of such a mechanism in the PNS is likely to suppress the subthreshold pain signals and/or termination of a nociceptive signaling response. We corroborate this notion by showing

increased *Pomc* gene expression in the DRG post capsaicin-induced acute pain in the hindpaws of healthy mice. Recently, the endogenous expression of many unlikely neurotransmitters and receptors, which can control peripheral nociception, is being reported in the DRG neurons. For instance, the expression of VEGFR1 by the sensory neurons promotes excitability[39],

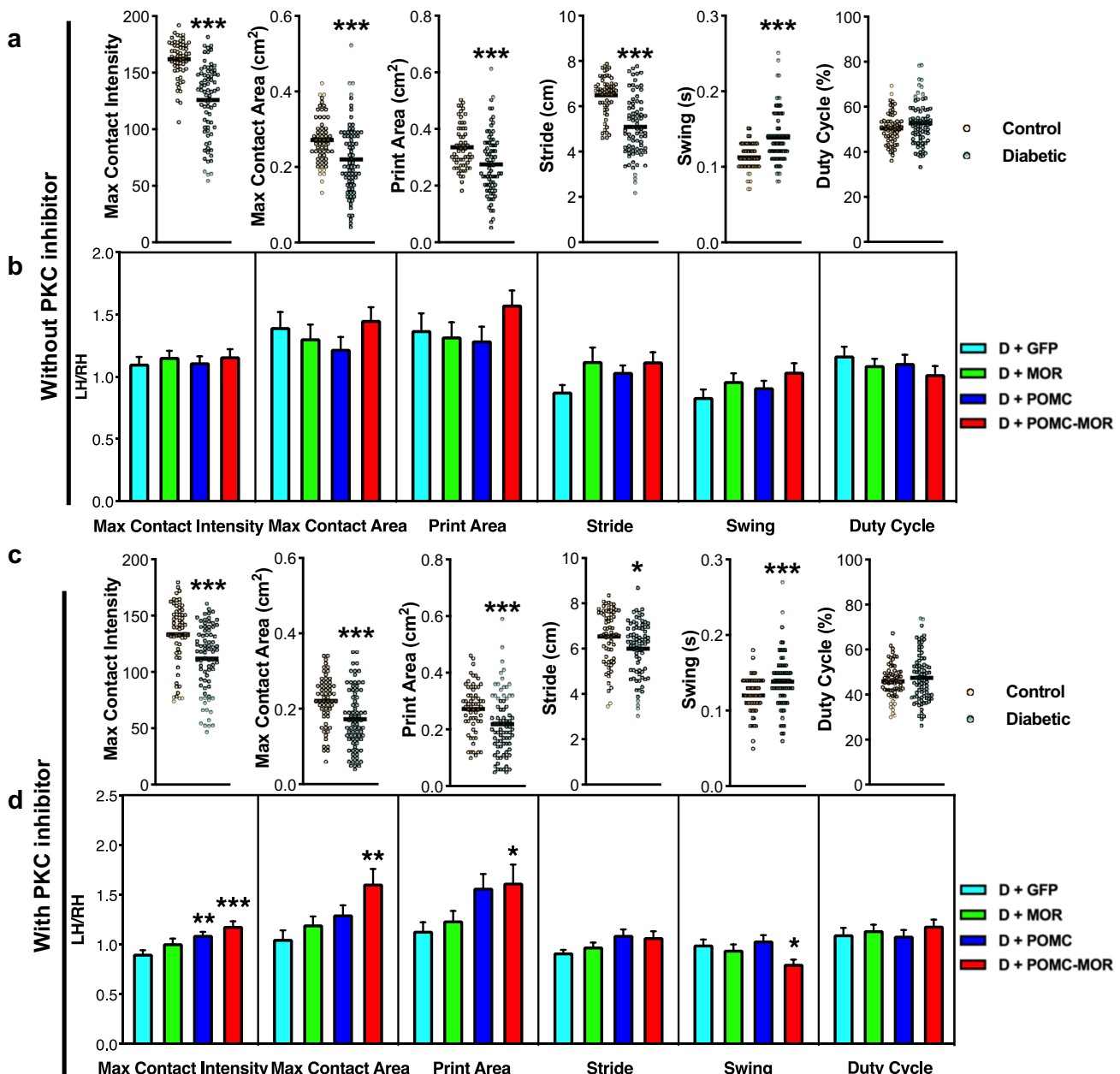

**Fig. 8 Genetic reconstitution of POMC and MOR improves gait parameters of diabetic mice.** Gait parameters analyzed using Catwalk system in naïve control and diabetic mice: **a** before and **c** after intraplantar PKC inhibitor injection, (Gö6983, 20 μM, 45 min) For panels (**a**) and (**c**), shown as dot plot with median; two-tailed *t*-test. Naïve control, *n* = 10, and Diabetic, *n* = 12; 5–6 readings per mouse. Circles represent individual data points. Orange circle: control, teal circle: diabetic mice. Gait parameters analyzed in diabetic mice overexpressing AAV-1/2 constructs in the right hindlimb: **b** before and **d** after intraplantar PKC inhibitor injection. Data expressed as a ratio of left hindpaw/right hindpaw. For panel (**b**), *n* = 10/cohort, 1–3 readings per mouse; one-way ANOVA with Dunnett's post-hoc test. For panel (**d**), D + GFP, *n* = 9; D + MOR, *n* = 10; D + POMC, *n* = 10; D + POMC-MOR, *n* = 10; 1–4 readings per mouse. For panel (**d**), **\***represents significant difference compared to D + GFP cohort. For panels (**b**) and (**d**), data represent mean ± SEM; one-way ANOVA with Dunnett's post-hoc test. Cyan bar: D + GFP, green bar: D + MOR, blue bar: D + POMC, red bar: D + POMC-MOR. For all panels, female mice data are shown; \**p* < 0.05, \*\**p* < 0.01, \*\*\**p* < 0.001 with 95% C.I. Source data are provided as a Source Data file.

whereas GABA is able to inhibit it[40]. We report an additional mechanism mediated by POMC that acts as a negative feedback mechanism for controlling the excitability of nociceptive sensory endings. This contributes to the ever-increasing role of DRG in the pain processing at a subspinal level, falling in line with the broadly accepted gate-control theory[41]. However, whilst POMC may participate to resolve acute pain under normal physiological conditions, this pathway is impaired during chronic condition such as diabetes. We demonstrate the importance of POMC-mediated antinociception using a mouse

model of DPN, in which the pain sensitivity increases as POMC levels drop in the DRG.

In the context of the POMC-pathway in diabetes, sensory ganglia are central to DPN, as are the metabolic changes occurring in them. A study reported about 15-fold increase in glucose, sorbitol, and fructose levels in the DRG of diabetic mice[42]. These were associated with significant changes in the expression of proteins involved in various pathways, such as, the LXR/RXR-pathway, which is known to affect POMC levels[43]. Furthermore, a recent study showed that targeting specific metabolic pathways in

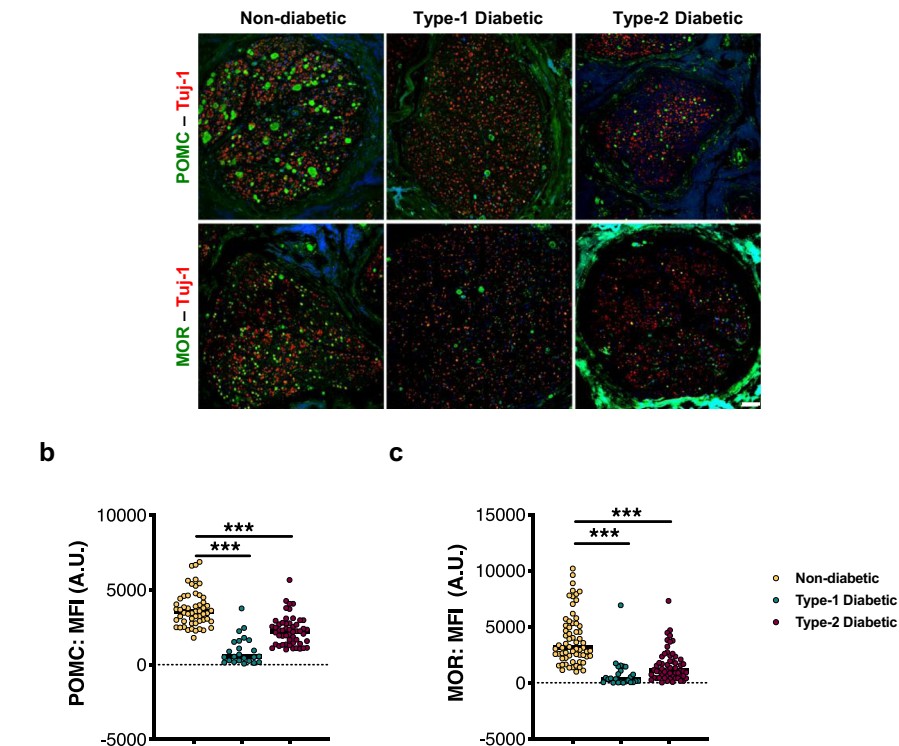

**Fig. 9 POMC and MOR are downregulated in the PNS of diabetic patients. a** Typical examples and quantitative analysis showing expression of: **b** POMC and **c** MOR proteins in Tuj-1+ sciatic nerve sections of human subjects, stained using specific antibodies (non-diabetic healthy control, $n = 11$; type-1 diabetic patients, $n = 4$ subjects and type-2 diabetic, $n = 11$). For panels (**b**) and (**c**), data represent dot plot with median; 5–7 sections from each subject; one-way ANOVA followed by Dunnett's post-hoc test; ***$p < 0.0001$ with 95% C.I. Each circle represents individual data point. Orange circle: non-diabetic, teal circle: type-1 diabetic and maroon circle: type-2 diabetic patient data. Scale = 50 μm. Source data are provided as a Source Data file.

the DRG neurons by inhibiting muscarinic receptors can reverse sensory anomalies during diabetes[44]. This accentuates the importance of the metabolic changes occurring in sensory ganglia with respect to peripheral nociception.

We show that during diabetes, high glucose-mediated NF-kB activation in the DRG causes *Pomc* promoter suppression. NF-kB has been a contested regulator of the *Pomc* promoter. In acute inflammatory conditions, such as an infection, NF-kB p65 subunit has been shown to increase the *Pomc* promoter activity[45,46], whereas the same subunit suppresses *Pomc* promoter under chronic low-grade inflammatory conditions[47]. A direct role for an inhibitory activity of NF-kB has also been described. It was shown that NF-kB directly binds *Pomc* promoter and suppresses CRH-mediated activation in AtT-20 cells and human melanocytes[48,49]. This is consistent with our findings, which demonstrate the inhibitory action of NF-kB on the *Pomc* promoter under hyperglycemic conditions, in vitro as well as in vivo. However, it has also been reported that high glucose can also lead to activation of the rat *Pomc* promoter by NF-kB in AtT-20 cells[50]. This discrepancy maybe dependent upon the exposure conditions, since NF-kB sites bind a larger variety of NF-kB subunit dimers.

The association of elevated NF-kB in the DRG and diabetic neuropathic pain has been previously demonstrated[51], thereby supporting our findings. It was shown that NF-kB p50$^{-/-}$ mice induced with STZ were resistant to tactile allodynia and that lowering NF-kB p50 levels specifically in the DRG improved painful DPN. Our data too show that the mechanism of *Pomc* promoter suppression in diabetes occurs via p50 homodimer, which have been shown to be transcriptionally repressive[52]. Another study demonstrated an increased activation of p65 subunit in the DRG of STZ-induced rats induces

hypersensitivity upon binding to the promoter of P2X3 receptor[53] (purinergic ion channel upregulated during neuropathic pain). The discordance in the subunit identification could be because of the different time-points of investigation post STZ-treatment, dosage of STZ, or animal species tested.

NF-kB-mediated POMC downregulation and consequent lowered ß-endorphin level, rendered one arm of this particular opioid signaling dysfunctional. Surprisingly, MOR protein expression was also decreased in diabetic mice. Excess of agonist can result in the phosphorylation, internalization, and degradation of MOR[17]. In the current scenario, agonist-independent MOR phosphorylation and degradation seemed to be the likely mechanism. PKC is reported to be responsible for MOR phosphorylation in the absence of an agonist[54]. Incidentally, hyperglycemia is known to cause a chronic activation of different PKC kinase isoforms and is closely linked in the pathogenesis of heart, kidney, eye, and nerve complications during diabetes[25]. PKC activation has only been measured in the sciatic nerves of diabetic animals and assumed to have an effect on nerve conduction velocities[25]. Our data demonstrate again that sensory ganglia are the seat of changes affecting the POMC-MOR axis, by showing an increased PKC activation and concomitant increase in MOR phosphorylation in the DRG during diabetes. PKC-mediated MOR phosphorylation followed by its internalization has been reported in the DRG of STZ-induced rats[55]. An internalized receptor can be recycled back to the cell surface. However, chronic activation of PKC would lead to persistent internalization leading to futile loop event. The impairment of successful recycling results in lysosomal targeting of MOR[17]. Our study not only shows PKC-mediated MOR internalization, but also an association between PKC activation and MOR lysosomal degradation in vivo under hyperglycemic conditions. Our finding of

MOR lysosomal degradation during diabetes is corroborated by Mousa et al.[56]

Besides phosphorylating MOR, PKC can also phosphorylate other potential targets known to be involved in pain transduction. For example, PKC-mediated phosphorylation of the excitatory transient receptor potential channels (e.g., TRPV1)[57,58] or voltage-gated sodium channels (Nav1.8)[59] in the DRG cause ion channel sensitization and is linked directly with heightened pain sensitivity. PKC phosphorylation of the inhibitory glycine receptors desensitizes the receptors, also resulting in increased neuronal excitability[60]. Moreover, PKC was also suggested to promote diabetic pain by the phosphorylation of purinergic receptors (P2X3) in the DRG of STZ-induced rats[61]. Viewed holistically, PKC activation is an important event capable of triggering several detrimental pathways contributing to DPN. Degradation of MOR is one of them.

The noteworthy observation was that diminished POMC and MOR levels in the sensory ganglia were associated with increased hypersensitivity towards evoked pain, as well as, with pain-related behavioral changes in the diabetic mice. We found that whilst POMC was necessary and sufficient to overcome the mechanical hyperalgesia, MOR aided the antinociceptive effect against thermal hyperalgesia and was indispensible in rectifying the altered gait of the diabetic mice. The antinociceptive effect of POMC in the absence of MOR could be explained by the fact that ß-endorphin is able to bind to other opioid receptors to some extent and trigger the downstream signaling[19,62]. However, MOR is essential to harness the full potential of this therapeutic strategy to overcome hyperalgesia.

This therapeutic intervention was effective in female and male diabetic mice at their respective peaks of hyperalgesia. We observed that the disease, in terms of pain sensitivity, progressed differently in both sexes over the course of time, with females tending towards hyperalgesia (12 weeks post-induction), whilst males displayed a brief hyperalgesic phase comparatively earlier (6 weeks post induction). Similar gender differences showing higher propensity of female sex to neuropathic pain and earlier onset of neuropathy in the males, have been previously reported in patients[63–67] and animal models of diabetic neuropathic pain[68,69]. To investigate why there is a gender-specific difference with respect to the DPN onset and progression is beyond the scope of this study. However, the underlying mechanism of POMC-MOR dysfunction occurred in both genders and coincided with heightened pain sensitivities. This would suggest that gender difference is not responsible for this downregulation.

Interestingly, our findings in type-1 diabetic mouse model were translatable to both type-1 and type-2 diabetic patient nerves of both genders, thereby increasing the credibility and scope of the proposed therapeutic target. Furthermore, this confirmed that downregulation of POMC and MOR during diabetes was not gender-specific. Unfortunately, we cannot present pain scores of the patients analyzed as this is a retrospective study and the nerves were obtained from a pre-existing tissue bank. Ethical reasons forbid us to obtain nerve material from living diabetic patients presenting symptoms with pain to correlate the molecular findings. This is a limitation, which may be overcome by planning a prospective long-term study with large number of participants. Despite the predicaments in obtaining patient nerves, we are able to present a snapshot view into the molecular events that verify our preclinical findings. Thus, our findings show the importance of DRG neuron-derived POMC and its antinociceptive signaling in the context of painful DPN.

There is a complex interplay of several aberrant mechanisms culminating in neuropathic pain during diabetes. Hyperexcitability of the sensory neurons (indicated by higher c-Fos expression during diabetes[70,71]) generating action potentials in the absence of a 'painful' stimulus is a major determinant underlying diabetic neuropathic pain[72,73]. This may occur two separate ways — Firstly, by the altered expression or modification of ion channels on the nociceptors that result in 'gain-of-function'. For example, increased expression of the Nav1.7[74] channels on the nociceptors has been reported in diabetes. Glycation of Nav1.8[75] and glycosylation of Cav3.2[76]ion channels leading to their increased activities during diabetes have also been reported previously. Another study showed increased activity of HCN2 (hyperpolarization-activated cyclic nucleotide-gated) channels as a result of increased cAMP levels in the DRG of diabetic mice[70]. We demonstrate that the underlying metabolic disturbances also cause the malfunction of the antinociceptive POMC-pathway, which constitutes the 'loss-of-function', and thereby promote a neuropathic pain state in diabetes. Reinstating this pathway helped overcome neuropathic pain, establishing the therapeutic significance of this inhibitory subspinal mechanism.

Studies reporting novel disease mechanisms are currently driving the momentum of developing pathogenesis-oriented therapeutics for painful DPN. For example, blocking the HCN2 channels using ivabradine[70] or scavenging methylglyoxal and preventing Nav1.8 glycation[75] have shown promise in the in vivo studies. Other emerging molecules include the selective sodium and calcium channel blockers, TRPA1 antagonists among others[77]. While most of these approaches target the excitatory ion channels/receptors, we propose a strategy of restoring body's natural defense mechanism of controlling pain signaling. It provides a platform for the development of therapies by boosting the endogenous POMC synthesis and prevention of MOR degradation. Administration of Pomc promoter agonists, such as CRH (routinely administered in the patients with Cushing's disease of POMC deficiency)[78] or melanocortin receptor agonists into the spinal cord may be one of the possibilities to enhance the ß-endorphin level in the PNS. Other compounds, such as the melanocortin receptor agonists, which have shown to enhance POMC expression in patients with genetic POMC deficiency[79,80], may be another promising option. Gene delivery techniques would also be a potential therapeutic option in the future for patients with painful DPN. Lastly, in light of the ongoing opioid crisis, our study offers a promising alternative approach to counter peripheral neuropathic pain.

## Methods

**Mice**. All animal experiments were approved by the local animal ethics committee (Regierungsspräsidien Karlsruhe, Germany). Mice were housed in groups of 2–4 mice/cage at the central animal facility of Heidelberg University under 12 h light–dark cycle with constant room temperature (~24 °C), humidity (~55%) and always had access to food and water.

*Diabetic mouse model*. Diabetes was induced in the 10 week old C57Bl/6 mice (Charles River). Streptozotocin (Sigma-Aldrich, 50 mg/kg) was administered i.p. on 6 subsequent days. Intradermal insulin injections were given bi-weekly as soon as the blood glucose levels reached 300 mg/dL[81]. Experiments were performed 12 weeks post-onset of hyperglycemia in female mice and 6 weeks post-onset in male mice.

*MOR$^{-/-}$ mice*. MOR-deficient mice were provided by Prof. Gaveriaux-Ruff. These mice were generated by crossing MOR-floxed mice with Cytomegalovirus-promoter-Cre (CMV-Cre) mice[82]. Oprm1-floxed mice were used as controls. Both mouse lines were on a mixed genetic background (50% C57BL6/J – 50% SV129Pas). Animals were produced in IGBMC (Institut de Génétique et de Biologie Moléculaire et Cellulaire, Illkirch, France). Experiments were performed on males and females aged 16–20 weeks as approved by the local body Com'Eth d'Ethique pour l'experimentation Animale IGBMC-ICS.

*Pomc conditional knockout mice*. The Pomc conditional knockout mice tissues were provided by Dr. Christoph Klose. These mice were generated by mating Pomc-floxed mice with the mouse line carrying neuron-specific cre driver (Snap25 cre). The Pomc-floxed mice were on the C56Bl/6 genetic background[83] and Snap25cre mouse line was backcrossed to C56BL/6 genetic background for several

generations. Experiments were performed on males and females aged 10–12 weeks as approved by the local body Landesamt für Gesundheit und Soziales Berlin (LAGeSo).

**Human tissue**. Human lumbar DRG and sciatic nerves were obtained with ethics approval from NBB (Netherlands Brain Bank, Amsterdam, Netherlands) and NCT (National Centre for Tumor diseases, Heidelberg, Germany), respectively. For patient details, refer to Supplementary Tables T2 and T10.

For examining POMC expression in human DRG, 6 healthy donors were selected (males, $n = 3$ and females, $n = 3$). To compare POMC and MOR expression under healthy and diabetic conditions, sciatic biopsies were selected from healthy ($n = 6$ males, $n = 6$, and $n = 5$ females, $n = 5$), type-1 ($n =$ males, $n = 4$, $n =$ females), and type-2 diabetic patients ($n = 6$ males, $n = 6$ and $n = 5$ females, $n = 5$).

**Plasmid constructs**. For in vitro transfection of AtT-20 cells, mouse *Pomc* promoter cloned in pGL3 vector was used (kind gift from Domenico Accili; Addgene plasmid # 17553). The NF-kB binding site in the promoter 'GGGAAGCCCC' was mutated to 'GGGAAGAACC' by site-directed mutagenesis. Primary DRG cells were transfected with mouse *Oprm1* cDNA (GE Dharmacon) cloned into the vector pmCherryN1 (Clonetech). For in vivo injections, recombinant AAV(1/2) particles were prepared using viral vector pAM-CBA-WPRE-bGH to clone either of the following mouse cDNA constructs: MOR-GFP, POMC-GFP, bicistronic POMC-2A-MOR-FLAG, or GFP alone as a control virus, while AAV(9/2) particles were prepared using the bicistronic POMC-2A-MOR-FLAG construct or only GFP. See Supplementary Table T1 for details on sequences used.

**Gene overexpression in STZ-induced mice**. Injection into the DRG: rAAV(1/2) particles were injected directly into the DRG of wild-type mice, as described previously[84]. Briefly, 8 week old mice were anesthetized with a cocktail containing dormicum/dormitol/fentanyl (3:8:2, 1 µg/g i.p.), 500 nl virus suspension was injected into the right L3 and L4 of the mice using a glass pipette and a microprocessor-controlled minipump (World Precision Instruments). Post surgery, each mouse was housed individually and administered with the same anesthetic cocktail subcutaneously to relieve pain. The mice were allowed to recover for 2 weeks before inducing them with STZ.

Intrathecal injection: Intrathecal injections were performed at 3, 4, and 5 weeks post STZ induction. The mice were injected with rAAV (9/2) particles as described in detail previously[85]. Briefly, the mice were anesthetized with 2% isoflourane and 20 µL virus suspension was injected using a Hamilton syringe into the groove of L5-L6 vertebrae. The injections were done two more times with a gap of 1 week for the mice to recover.

Then, 10–12 mice were allocated randomly to each group receiving a specific virus by an investigator blind to the virus identity. Behavioral testing was performed 12 weeks post STZ induction. Each mouse was then sacrificed to monitor viral spread by assessing GFP or FLAG fluorescence in the lumbar DRG. Mice showing less than 20% infected neurons were excluded from the analysis. A detailed characterization of rate of neuron transduction and neuron-specificity for both AAV (1/2) and AAV (9/2) in mouse DRG has been previously described[32,86].

**Cell culture and transfection**. AtT-20 cells (Sigma-Aldrich) were cultured in DMEM and 10% FCS. They were transfected using Neon electroporator at 1200 V, 20 ms, and 2 pulses. Transfected cells were harvested 48 h later.

*Primary DRG culture*. DRG were isolated from 4–6 week old female C57Bl/6 mice[30], 30–40 DRG were collected from each animal and digested using 0.3% collagenase and 0.05% trypsin-EDTA at 37 °C. The digested tissue was passed through glass pipette to get an even cell suspension. The cells were washed in PBS, resuspended in buffer R (Thermo Fischer), and equally divided into number tubes required for exposure conditions; then 1 µg of MOR-mCherry construct was added per tube except for the 'untransfected control'. The cells were then transfected using the Neon electroporator system at 1300 V, 20 ms, and 2 pulses. The transfected cells were seeded on poly-L-ornithine- and laminin-coated coverslips. Selection medium specific for neuronal cells, containing AraC (5 µM) and NGF (10 ng/mL), was added 12 h post transfection. After 24 h of transfection, the cells were incubated with either of the following medium: normal glucose medium (17 mM), normal glucose + PKC inhibitor (Gö6983, 1 µM), high glucose (40 mM), or high glucose + PKC inhibitor (Gö6983, 1 µM). The cells were stained and analyzed microscopically 48 h post exposure.

**Immunocytochemistry**. Primary DRG cells were stained live with wheat germ agglutinin (WGA, 5 µg/mL) for 10 min at 37 °C and then fixed with 4% paraformaldehyde. The cells were stained with DAPI, mounted onto slides, and imaged using confocal laser-scanning microscope, 63X objective (LSM800, Zen2 software, Zeiss). Surface association of MOR was calculated by overlap of WGA and mCherry signal using ImageJ software.

**Nociceptive tests**. All animal experiments were approved by the local governing body for animal ethics (Regierungspräsidien Karlsruhe, Germany). Behavioral tests were performed in awake, unrestrained, and age-matched mice during the day (light phase) by two investigators in a double-blind manner.

*Capsaicin test*. Capsaicin (Cat. # 0462, Tocris) was injected intraplantar as a 0.03% solution in PBS in a volume of 20 µL and nocifensive behaviors, such as licking, shaking, or flicking of the injected paw, was measured over the time period of 5 min after injection[87].

*Thermal and mechanical hyperalgesia*. Thermal and mechanical hyperalgesia were measured in the mice using Hargreaves[88] and von Fray filaments[89], respectively. Prior to each assay, the mice were acclimatized to the experimental setup (Ugo Basile) several times. For the Hargreaves test, a ray of infrared light was directed to the plantar surface of each hindpaw and the paw withdrawal latency (PWL) was measured. Each paw was measured 5 times with an interval of 10 min between consecutive readings.

For mechanical sensitivity measurement, von Frey monofilament with the bending forces 0.16 g, 0.4 g, 0.6 g, 1.4 g, and 2 g were applied 5 times onto the plantar skin overlying the calcaneus bone of each paw. Readings were recorded 5 times per paw per filament. Data are plotted as paw withdrawal response (PWR) calculated as the percent of positive responses per filament per mouse.

A set of readings was also taken 45–60 min after intraplantar injection of PKC inhibitor (Gö6983, 20 µM).

**Place escape/avoidance paradigm (PEAP)**. PEAP was conducted using a modified von Frey setup[90]. The PEAP apparatus consisted of a plexiglass box, whose one half was dark and the other half was light, using adhesive tapes of black and white colors, respectively. The two halves were separated by a barrier, which allowed free movement of the mice between dark and light chambers. The box was placed on the von Frey wire mesh and readings were taken in a dark room except for the light coming from the bulb placed on the light chamber. The mice were placed in this apparatus for 30 min comprising of an initial 5 min of exploratory phase, followed 25 min of testing phase. In the dark chamber, the left (or contralateral) hindpaw was stimulated, while in the light chamber the right (or ipsilateral) hindpaw was stimulated using 0.6 g Frey filament. The time spent in the light chamber was measured for every 5 min interval. A set of readings was also taken 45–60 min after intraplantar injection of PKC inhibitor (Gö6983, 20 µM).

**Gait analysis**. Gait analysis was conducted using the Catwalk XT version 10.6 (Nodulus)[91]. The Catwalk apparatus consists of a 1.3 m long enclosed black corridor on a glass plate, through which the mouse is allowed to traverse freely from one end to the other. As the mouse walks, its paws are captured by a high-speed video camera using illuminated footprints technology. Footprint classification and error correction are carried out by the system automatically. The mice were habituated to the apparatus several times prior to each assay. For each mouse analyzed, a set of readings was taken without and with the intraplantar injection of PKC inhibitor (Gö6983, 20 µM, 45–60 min post injection). Each mouse was allowed to cross the corridor thrice. Gait analysis was performed for hindpaws using the following parameters: Paw print area, Maximal contact intensity of a paw, Maximum contact area, Swing, Stride, and Duty cycle. See Supplementary Methods for detailed information.

**Immunofluorescence assay**. Lumbar DRG and sciatic nerves were dissected from the mice after perfusion with ice-cold PBS. The tissues were fixated in 4% paraformaldehyde, cryoprotected, and cryosectioned at 10 µm. Every fifth consecutive section from the tissues was used for each staining. The sections were acetone-fixated and washed with 50 mM glycine for antigen retrieval; 0.5% saponin was used to permeabilize and 10% donkey serum in PBS-0.2% Triton X100 for blocking. Following an overnight incubation with the primary antibodies (Supplementary Table T12) diluted in blocking buffer, the sections were washed and incubated with respective fluorochrome-conjugated secondary antibodies. The sections were finally stained with DAPI and mounted with Permaflour mountant medium (Thermo Fischer).

Human DRG and sciatic nerves were fixated in formalin and embedded in paraffin, then 5 µm thick sections were cut. Citrate buffer (pH = 6) was used for antigen retrieval. Immunostaining was performed in the same manner described above. Fluorescent images were taken on confocal laser-scanning microscope (LSM700 or LSM 800, Zen2 software, Zeiss) and analyzed using ImageJ software.

**EIA**. From each mouse, 12 lumbar DRG, 2 sciatic nerves, and 2 footpads were dissected, snap-frozen, and crushed using frozen glass pestle. Crushed tissue was incubated with 1 M acetic acid at room temperature and then at 95 °C, 15 min each. The samples were cooled on ice and centrifuged (4000 g, 15 min, 4 °C). The supernatant containing ß-endorphin was collected and incubated with avidin-coated magnetic beads to preclear the supernatant of endogenous biotin. The precleared supernatants were then lyophilized and resuspended in 1X assay buffer of the ß-endorphin EIA kit (Phoenix Pharmaceuticals). EIA was carried out according to the manufacturer's instructions. The sample absorbance values

obtained were plotted on a standard graph constructed using known concentrations of ß-endorphin (FLUOmega software). The concentration of peptide calculated for the samples was adjusted according to quantity of protein added per well and ß-endorphin concentration per mg or protein was determined for each sample.

**ChIP**. Snap-frozen lumbar DRG were gently thawed on ice and incubated with 18.5% formaldehyde for protein–DNA crosslinking. After 10 min, glycine was added to a final concentration of 0.125 M. The DRG were mechanically dissociated and centrifuged (800 $g$, 5 min, 4 °C). The nuclear pellet was incubated with micrococcal nuclease (Cell Signaling) in MN buffer (10 mM TrisCl pH=7.5, 4 mM $MgCl_2$, 1 mM $CaCl_2$, PIC) at 37 °C for 10 min. The samples were centrifuged (10,000 $g$, 1 min, 4 °C) and chromatin-containing pellet was sonicated (20 s, 3 pulses). The samples were precleared using protein A/G magnetic beads. Then, 60 µg chromatin from each sample was incubated overnight at 4 °C with 1 µg of an appropriate NF-kB subunit antibody (Supplementary Table T12) along with magnetic protein A/G beads. The rest of the procedure was carried out using ChIP Magna kit as per manufacturer's instructions (EMD, Millipore). The quantification of *Pomc* promoter fragments specific for NF-kB binding sites were quantified using specific primers (Supplementary Table T1) using a quantitative real-time PCR.

For ChIP analysis on AtT-20 cells, cells were exposed to either 5 mM or 20 mM glucose-containing medium with reduced serum. Post 12 h of exposure, the cells were cross-linked with formaldehyde to a final concentration of 1%. The rest of the procedure was same as above.

**RNA extraction and qRT-PCR**. RNA was isolated from snap-frozen lumbar DRG using the Trizol method (Invitrogen) and purified using DNAseI treatment (Invitrogen) as per manufacturer's instructions. For cDNA synthesis, 1 µg of total RNA was diluted in cDNA Master Mix using reagents from cDNA synthesis kit (Applied Biosystems). Relative expression of target genes was measured by qRT-PCR and normalized with the expression of 18srRNA, using the Lightcycler 480 real-time PCR system (Roche). Primer sequences are given in Supplementary Table T1.

**Electromobility shift assay (EMSA)**. AtT-20 nuclear lysates were incubated with a freshly prepared radioactively labeled DNA probe (66.7 nM), consisting of NF-kB consensus sequence (Supplementary Table T1) along with a reaction mix containing 0.1 mg/mL BSA, Buffer D (20 mM Hepes-KOH; pH=7.9, 20% Glycerol, 100 mM KCl, 0.25% NP40, 0.5 mM EDTA, 2 mM DTT), and Buffer F (100 mM Hepes-KOH; pH=7.6, 20% Ficoll 400, 300 mM KCl, 10 mM DTT) for 30 min at room temperature. For supershift EMSA, the nuclear lysates were incubated with 1 µg of antibodies for the NF-kB subunits p65, p52, p50, and cRel on ice for 20 min before incubating with the reaction mix. Reactions were then loaded onto a pre-electrophoresed 6% acrylamide/bis (37.5:1) gel in 0.5xTBE and run at 100 V. The gels were dried and analyzed by exposing them to photographic film.

**Immunoprecipitation**. Immunoprecipitation of MOR from total DRG lysates was carried out using MOR (UMB3) antibody cross-linked to Protein A-conjugated magnetic beads. Rabbit IgG-conjugated beads were used as negative controls. The antibody cross-linked beads were added to lumbar DRG lysates (containing phosphatase inhibitor) pooled from 3 control or diabetic mice and incubated overnight at 4 °C. The beads were washed with IP buffer and bound proteins were eluted using 1 M glycine (pH = 2.5). The elution step was repeated thrice and tris-HCl (pH = 10) was added to the eluent to stabilize the pH. The immunoprecipitated protein was loaded in equal amounts onto 4–20% tris glycine SDS-PAGE gel and blotted onto a nitrocellulose membrane. The membrane was probed with pan-phosphothreonine antibody to detect the phosphorylation of MOR. A MOR antibody (#AOR-011) was used as a loading control. See Supplementary Table T12 for antibody details.

**Immunoblotting**. Snap-frozen tissues were then mechanically homogenized in ice-cold RIPA buffer, incubated on ice for 1 h and then centrifuged at 20,000 $g$ for 30 min. The supernatant was collected and electrophoresed on 4–20% tris glycine SDS-PAGE gel. The proteins were blotted on nitrocellulose membrane and blocked using non-protein blocking buffer (Thermo Fischer) and incubated overnight at 4 °C with primary antibodies (Supplementary table T12). The membrane was washed and incubated with an appropriate HRP-conjugated secondary antibody. The signal was developed using ECL reagent (GE Healthcare).

Nuclear lysates from AtT-20 cells were prepared as follows: The cells were pelleted and resuspended in Buffer A (10 mM Hepes-KOH; pH=7.6, 1 mM MgCl2, 10 mM KCL, 0.5 mM DTT), and incubated on ice for 10 min. The nuclei were pelleted by centrifugation (20,000 $g$, 5 min, 4 °C) and resuspended in Buffer C (20 mM Hepes-KOH; pH=7.9, 25% Glycerol, 420 mM NaCl, 1.5 mM MgCl2, 0.5 mM EDTA, 0.5 mM DTT). The suspension was further incubated on ice for 20 min and finally centrifuged (20,000 $g$, 5 min, 4 °C) to pellet the debris. The supernatant containing nuclear proteins was collected.

Then, 6–9 samples from independent experiments were analyzed by densitometry using ImageJ and normalized with β-actin (total lysate) or histone H3 (nuclear lysates).

**Statistics and reproducibility**. An unpaired (two-tailed) Student's $t$-test was used for comparing two groups for a single parameter at a single time-point. For multiple comparisons in multiple groups for random or repeated measures, analysis of variance (ANOVA) was employed to measure multiple treatments compared with a control group. Appropriate post-hoc tests recommended by the Graphpad Prism software were applied to determine statistically significant difference. $p$ values less than 0.05 were considered significant. For reproducibility, each experiment was repeated independently 3–20 times with similar results.

## Data availability
The authors declare that all other relevant data are available from the authors upon reasonable request. Source data are provided with this paper.

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

## Acknowledgements

We are grateful to Serap Kaymak, Axel Erhardt, Nadine Gehrig, and Elizabeth Kliemank for their excellent technical assistance. We specially thank Luisa Fuchs for her contribution to perform revision experiments and Dr. Andreas Fischer for his support to perform the imaging experiments. We are also thankful to the Deutsche Forschungsgemeinshcaft for funding this project via the DIAMICOM graduate school, SFB1158 (project A03, B01 and B06) and SFB1118 (project A04 and B06). The study was additionally supported by the European Union's Seventh Framework Programme (FP7/2007-2013) under grant agreement No. 602919 (CGR). We are thankful to Interdisciplinary Neurobehavioral Core (INF 515, 69120 Heidelberg) for providing facilities to conduct behavioral tests and to DKFZ Light Microscopy Core Facility.

## Author contributions

D.D. designed and performed the experiments, analyzed the data and wrote the manuscript. T.F. supervised the study and helped with manuscript writing. N.A. helped with virus production and DRG injections. C.G.R. provided MOR$^{-/-}$ and contributed to manuscript writing. C.K. provided *Pomc* conditional knockout mice, facilitated revision experiments and contributed to manuscript writing. A.T.T. helped with revision experiments. R.K. provided invaluable conceptual input, helped write the manuscript and provided all facilities to carry out virus-related procedures, surgeries and behavior tests. P.P.N. conceived, designed and supervised the study.

## Funding

## Competing interests

The authors declare no competing interests.
