## [Peer Review File · Nature Communications]

Reviewers' comments:

Reviewer #1 (Remarks to the Author):

The authors show that proopiomelanocortin (POMC), an endogenous opioid precursor peptide, is dysfunctional in the sensory neurons in organisms with diabetes.

They show loss in the basal POMC levels in peripheral nerves of streptozotocin-induced diabetic mice and in tissue from diabetic patients, and a further impairment of POMC mediated antinociception by lysosomal degradation of μ -opioid receptor (MOR). Finally, they rescue the phenotype by viral overexpression of POMC and MOR in the sensory ganglia.

This very well written paper about a very well performed study shows a new mechanism by which diabetic neuropathy causes pain, and a new target of pain treatment in diabetes. While this is intriguing, the question arises about the relative importance of the POMC pathway. The authors themselves have previously published methylglyoxal modification of the sodium channel Nav 1.8 as a major component causing pain in diabetes. Others have shown further peripheral and central pathways. So where do the present findings fit in, and what is their relative importance?

Are the findings related to diabetes or to nerve lesion as such? I.e., would the same findings occur in a model of nerve injury or chemotherapy induced painful neuropathy?

Another major question is the point about hyperalgesia only occurring in female mice. If this is so, and if the POMC pathway is very important for hyperalgesia, the downstream findings should be different in male mice. If the biochemistry is the same in male and female mice, and only female mice develop hyperalgesia, this casts some doubt on the causative connection. Furthermore, if the mechanism is only valid in females, the human data and the importance for male and female human diabetic patients should be discussed.

Reviewer #2 (Remarks to the Author):

This is a well-crafted story, reporting that sensory DRG neurons express POMC in mice and human and suggesting its antinociceptive role. POMC is expressed in multiple organs and different cell types, including neurons in the hypothalamus, cells in the pituitary gland, melanocytes in the skin. So far, very little is known about POMC expression in the peripheral nervous system.

The authors do a very nice job demonstrating the expression of POMC in DRG in both mice and human. Moreover, the authors find that POMC-MOR axis' expression is downregulated in both the diabetic animal model and patients. This finding is very novel and interesting. Furthermore, the authors find that reconstitution of POMC in DRG rescues both thermal and mechanical hyperalgesia, suggesting its importance and clinic significance of POMC in Painful neuropathy. Although it is known MOR exists in DRG, its local endogenous ligands are not clear. Thus, this manuscript makes a substantial contribution to literature, as well as extending our understanding of PNS POMC in the control of pain sensitivity.

Major concerns:

1. The major weakness is that given that POMC-MOR expression is downregulated in both female and male animals, why female diabetic mice are hyperalgesia while and male diabetic mice are hypoalgesia. Is there any difference between male and female in DRG POMC expression? If overexpression POMC in DRG in normal or diabetic male mice, will this also alter the pain sensitivity?
2. Since POMC expression is downregulated under diabetic conditions, it is also interesting to know whether POMC expression will be increased in response to acute pain stimulus. That is whether DRG POMC plays an antinociceptive role under normal conditions.

Minor points:

1. Fig.1b, POMC immunostaining is weak and almost covered by the red channel, which make it difficult to see POMC expression clearly.
2. Fig.3c, in the control panel, it seems that nearly all Tuj-1neurons express MOR in this picture, however the quantitative data shows only 60% neurons are MOR positive. The contrast setting of images may result in statistical errors. Can the authors provide raw confocal image files? Similar concern in Fig. 2d and Fig.s2.
3. Regarding to the specificity of POMC immunostaining in DRG, can the authors provide a positive/negative control of the antibody they used? For example, using the hypothalamic arcuate nuclei or the pituitary gland as a positive control and other brain areas as a negative control.

Reviewer #3 (Remarks to the Author):

In this work, Deshpande and colleagues have identified a pain-promoting pathway which involves a reduction of proopiomelanocortin (POMC) signaling (at both the ligand and receptor levels) in hyperglycaemic conditions encountered in diabetic neuropathy. The authors trace back these changes to the periphery, namely the dorsal root ganglion (DRG) and associated axons innervating the skin. Rescuing the diabetes-induced downregulation of POMC signalling via viral overexpression of the ligand and/or its receptor in the DRG provided analgesic effects across a variety of pain modalities. The authors propose this mechanism could form the basis for new antinociceptive therapies in humans.

The study contains a significant amount of data, including extensive behavioural pain testing. The existing experiments are conducted carefully using all the necessary controls and the results are explained in detail. The addition of human samples further substantiates the data obtained in mice. The methods are written in sufficient detail to allow replication of the experiments. The statistical analysis employed for each comparison is denoted clearly and is appropriate. Overall, the paper is very well written – experiments follow a logical flow and the narrative is clear. The conclusions are supported by the experimental data and the findings are novel.

While the results are interesting and solid, there is some key information missing which could extend the applicability of the conclusions and make it more translational.

1) The authors focused on female diabetic mice on the basis that this sex develops thermal hypersensitivity (as opposed to hyposensitivity in males) at the examined time points after diabetes induction. Given the recently emerged knowledge on sexual dimorphism in pain responses, it is important to characterise this pathway a bit more with regards to sex. Obvious questions are:

- Does the downregulation of POMC and MOR occur in the same way in male diabetic mice?
- Does viral overexpression of POMC/MOR reverse pain behaviours in male diabetic mice? This may not be possible to assess for thermal hypersensitivity, but it is definitely an option for mechanical pain as well as spontaneous pain tests.

2) The authors conclude that the site of interest is the periphery where they detect downregulation of the ligand/receptor. The evidence for this seems clear. However we cannot exclude an additional site of action in the CNS, where MOR is also expressed. Is there evidence for downregulation of this pathway in the spinal cord? Viral overexpression in DRG has antinociceptive effects but can the authors exclude that a) ligand is transported to central terminals to act on postsynaptic MOR in the spinal cord b) the injected AAV does not transduce spinal cord neurons?

3) It would be very interesting to ascertain whether this pathway is specific to metabolic neuropathies such as diabetic neuropathy, or a more generalised mechanism that also extends to traumatic

neuropathies. The authors could use a model such as spinal nerve ligation to quickly check whether POMC/MOR downregulation also occurs in the context.

4) While the PKC-mediated degradation of MOR is a plausible mechanism, there could be other pathways at play eg phosphorylation of ion channels. The authors should acknowledge this possibility in the discussion. Do PKA inhibitors also have an effect on MOR internalisation and/or diabetic pain?

Some more minor points:

While the authors provide a control staining (pre-incubation with antigen) to demonstrate the specificity of the POMC antibody, unfortunately this control is of limited use as it only shows that the antibody correctly recognises this antigen – it is however still possible that it also recognises distinct but similar proteins. The ideal control would be testing the antibody on tissue taken from POMC KO mice, which appear to be available (e.g. <https://doi.org/10.1073/pnas.0306931101> and <https://www.jax.org/strain/003191>). If the authors get some of these mice, it would also be very interesting to check their baseline pain responses as well as how they develop diabetic pain.

The presented work identifies a pain-protective pathway that could be exploited for treatment of diabetic pain, but presumably the POMC/MOR downregulation is not the source of diabetic pain. It would be useful to have a very brief discussion of established and emerging candidate molecules that could be responsible for the pathogenesis of pain such as Nav1.8, HCN2, TRPV1, Cav, especially since there might be overlap of pathways involving cAMP/PKA/PKC (eg <https://stm.sciencemag.org/content/9/409/eaam6072>, <https://www.nature.com/articles/nm.2750>, <https://www.ncbi.nlm.nih.gov/pmc/articles/PMC2735619/>)

In terms of applicability of results in the clinic, the authors could perhaps extend the discussion. How are these results likely to change the already established use of opioids for pain control in diabetes? Since we need to restore expression of ligand/receptor, would the proposed way forward be introducing copies of the protein via gene therapy?

Deshpande et al.

Loss of POMC-mediated antinociception contributes to painful diabetic neuropathy

We are grateful to the editor and all of the reviewers for their encouraging and constructive comments for our initial manuscript. We have now performed a series of new experiments to address each comment successfully. The reviewers' suggestions have helped improve the study and expand the scope of the findings. Below, we have presented our new data in a point-by-point response. The newly added text is highlighted in the revised version of the manuscript.

Reviewers' comments:

Reviewer #1 (Remarks to the Author): *The authors show that proopiomelanocortin (POMC), an endogenous opioid precursor peptide, is dysfunctional in the sensory neurons in organisms with diabetes. They show loss in the basal POMC levels in peripheral nerves of streptozotocin-induced diabetic mice and in tissue from diabetic patients, and a further impairment of POMC mediated antinociception by lysosomal degradation of μ -opioid receptor (MOR). Finally, they rescue the phenotype by viral overexpression of POMC and MOR in the sensory ganglia.*

Reviewer 1, Comment 1: *This very well written paper about a very well performed study shows a new mechanism by which diabetic neuropathy causes pain, and a new target of pain treatment in diabetes. While this is intriguing, the question arises about the relative importance of the POMC pathway. The authors themselves have previously published methylglyoxal modification of the sodium channel Nav 1.8 as a major component causing pain in diabetes. Others have shown further peripheral and central pathways. So where do the present findings fit in, and what is their relative importance?*

Author response: We thank the reviewer for the positive comments.

The reviewer raises a valid conceptual point regarding the importance of the POMC pathway in the holistic picture of painful diabetic peripheral neuropathy (DPN), which **we discuss below and have included this important point in the Discussion section of the Revised Manuscript on pg 21.**

DPN can be viewed as a disease consisting of a complex interplay of several aberrant mechanisms ultimately resulting in neuropathic pain during diabetes. Hyperexcitability of the injured sensory neurons generating action potentials in the absence of a 'painful' stimulus, is a major determinant underlying diabetic neuropathic pain^{1,2}. At the subspinal level, this hyperexcitability may occur in two separate ways: Firstly, the altered expression or modification of ion channels on the nociceptors that results in '*gain-of-function*', culminating in increased neuronal excitability. For instance, an increased expression of Na_v1.8 (ion channel amplifying action potential in neurons) in the sensory neurons of diabetic animal model has been reported³. Increased activity of Na_v1.8⁴ and Ca_v3.2⁵ due to glycation and glycosylation respectively has also been shown to promote painful DPN. Phosphorylation of Na_v1.8 by protein kinase C (PKC) is yet another mechanism known to underlie hypersensitivity in animal models of neuropathic pain⁶. We demonstrate increased levels of

PKC in the DRG of diabetic mice (**Revised Manuscript Figure 5a**), which may contribute to increased Na_v1.8 activity linked with painful DPN.

We propose that the impaired POMC antinociceptive pathway constitutes the '*loss-of-function*', which also promotes neuropathic pain phenotype. For the termination of a nociceptive response, inhibitory signalling is vital, the dysfunction of which can lead to persistent generation of action potential and result in hyperexcitability. This is shown to occur during diabetes due to reduced expression and activity of the inhibitory potassium ion channels K_v an animal model of diabetes⁷.

The summation of the nociceptive inputs from the PNS neurons influences synaptic transmission within the spinal cord. Therefore, enhanced input from the persistently active nociceptors further amplifies the nociceptive signalling leading to central sensitization, a phenomenon observed during painful DPN⁸. Downregulated POMC-mediated antinociception may exacerbate the local functional imbalance between facilitation and inhibition of the nociceptive signalling, which can collectively influence the ascending pain pathways during diabetes, a theme discussed in greater detail by Marshall A *et al*⁹, Tesfaye *et al*¹⁰ among others.

The dysregulation of the POMC pathway, observed in this study, co-exists with other aberrant pathways, which would together promote hypersensitivity in DPN. However, we showed that reinstating the POMC-mediated antinociception helps overcome painful DPN demonstrating the importance of this inhibitory subspinal mechanism.

Reviewer 1, Comment 2: *Are the findings related to diabetes or to nerve lesion as such? I.e., would the same findings occur in a model of nerve injury or chemotherapy induced painful neuropathy?*

Author response: We thank the reviewer for raising this crucial question.

According to our findings, the downregulation of POMC occurs in diabetes via NF-κB mediated promoter suppression, while elevated PKC levels underlie MOR degradation. Both NF-κB and PKC activation have been shown to be a consequence of metabolically-triggered changes during diabetes¹¹.

For the current study, we employed a low-dose Streptozotocin (STZ) model, which does not cause direct neurotoxicity¹². Using this mouse model, our group recently published a study on extensive characterization of morphological changes in the peripheral nerves of STZ-induced mice, using a high-resolution magnetic resonance neurography. This study showed that there were no focal lesions in the peripheral nerves of the diabetic mice. The first occurrence of subtle ultrastructural changes were observed no earlier than 24 weeks post STZ-induction¹³. These changes were associated with a hypoalgesic phenotype in the diabetic mice. In the current study, our observations are made at no later than 12 weeks post STZ-induction and correlate with a hyperalgesic phenotype, making their association with metabolically triggered changes stronger than the ultrastructural changes in the PNS of our diabetic mouse model.

In order to answer the second part of the question, we performed molecular analysis of the DRG of mice with spared nerve injury (SNI)¹⁴. We measured POMC and MOR protein levels in the L3 and L4 DRG of the ipsilateral and contralateral side of SNI and sham-operated mice at 7 days post-operation. No changes were observed in the DRG of the sham-operated mice.

Surprisingly, POMC and MOR were both downregulated in the ipsilateral DRG of SNI mice (**Reviewer Figure 1**). The downregulation of MOR in the DRG after SNI has been previously reported by Wieskopf *et al*¹⁵. However, our preliminary findings show that the malfunction exists also upstream of the receptor, *i.e.* at the ligand level. To investigate the mechanisms underlying as the downregulation of POMC-MOR in SNI, is beyond the scope of this study. The well-studied chronic neuroinflammatory changes (including NF- κ B¹⁶⁻¹⁸ and PKC^{19,20} activation) occurring in the DRG during SNI could be a contributing factor. These findings present an interesting basis for future studies with different focus.

Taken together, our preliminary findings in the SNI mouse model indicate that the POMC-pathway may represent a more generalized mechanism and possibly be extended to other chronic neuroinflammatory conditions.

Reviewer Figure 1: POMC and MOR protein levels are decreased the DRG of SNI mice (7d post-surgery). Representative blots and densitometric quantification of POMC (~ 26KDa band normalized to actin) and MOR protein (~ 55KDa band normalized to actin) in total lysates of L3 and L4 DRG (n=4-5 per group; two tailed t-test). Data represents mean \pm SEM. *p<0.05, **p<0.01, ***p<0.001.

Reviewer 1, Comment 3: Another major question is the point about hyperalgesia only occurring in female mice. If this is so, and if the POMC pathway is very important for hyperalgesia, the downstream findings should be different in male mice. If the biochemistry is the same in male and female mice, and only female mice develop hyperalgesia, this casts some doubt on the causative connection. Furthermore, if the mechanism is only valid in females, the human data and the importance for male and female human diabetic patients should be discussed.

Author response: We agree with the issue raised by the reviewer. To satisfactorily clarify this concern regarding the gender differences, we carried out extensive characterization of male diabetic mice.

In our initial manuscript, to determine the time point of maximum hypersensitivity, the male mice were tested for thermal hyperalgesia using Hotplate, which is a relatively broad measure. In our revised manuscript, we performed a thorough pain phenotyping over the course of time.

We examined the male mice for mechanical hyperalgesia (using Von Frey filaments) and thermal hyperalgesia (using Hargreaves) every 2 weeks post STZ-induction and compared them with age-matched healthy controls. In our setting using a low-dose STZ protocol, a brief thermal hypersensitivity phase was detectable using Hargreaves at 6 weeks, after which the male diabetic mice tended towards hypoalgesia (**Reviewer Figure 2a, Revised Manuscript Figure S5**). This small but significant difference in thermal hypersensitivity was detectable using Hargreaves, but not in the Hotplate method. The male mice displayed a comparatively prolonged period of mechanical hypersensitivity with a peak at 6 weeks post-induction (**Reviewer Figure 2b, Revised Manuscript Figure S7a**). The pain phenotyping we conducted for our initial manuscript was at 4, 8 and 12 weeks post-induction. We therefore had missed this brief phase of hyperalgesia at 6 weeks detectable using Hargreaves and von Frey filaments.

Having identified hypersensitivity peak at 6 weeks post-induction, we harvested the DRG and quantified POMC and MOR protein levels in the diabetic male mice. Consistent with our findings in the female diabetic mice, the POMC and MOR protein levels were significantly decreased in the DRG of male diabetic mice at this time point (**Reviewer Figure 2c, Revised Manuscript Figure S7**).

In terms of pain sensitivity, the disease progressed differently in both sexes over the course of time, with females tending towards hyperalgesia at 12 week post-induction, whilst the males displayed a hyperalgesic phase comparatively earlier. Though both genders displayed a neuropathic pain phenotype, we noted that the hypersensitivity (thermal and mechanical) was more pronounced in the female diabetic mice than in the male diabetic mice (**Reviewer Figure 3, Revised Manuscript Figure 2a, 2b, S7a, S7b**). Similar gender differences showing higher propensity of female sex to neuropathic pain and earlier onset of neuropathy in the males, have been previously reported in patients²¹⁻²⁵ and animal models of diabetic neuropathic pain^{26,27}. To investigate why there is a gender-specific difference with respect to the DPN onset and progression is beyond the scope of this study. However, the key finding is that when both male and female mice display the hypersensitivity, there is a concurrent loss of POMC and MOR regardless of the gender. This would suggest that gender difference is not responsible for this downregulation. This is further supported by the observation that the decrease of POMC or MOR was observed in both, female and male patients. This issue has been addressed on **pg 20 of the Revised Manuscript**.

In addition, we were also able to show that overexpression of POMC-MOR in the male diabetic mice ameliorated thermal and mechanical hypersensitivity, as well as, the gait anomalies (**Reviewer Figure 4, Revised Manuscript Figure S11, S12 and S13**). This further demonstrated that the antinociceptive effect of the proposed therapeutic strategy is not gender-specific.

Reviewer Figure 2: POMC and MOR levels are decreased in the DRG of male diabetic mice showing neuropathic pain phenotype . a Thermal and **b** mechanical hyperalgesia measured in male mice after 4 (n=8), 6 (n=8), 8 (n=8), 10 (n=8) and 12 (n=8) weeks post STZ-induction. For each time point, values normalized and statistically compared with those of age-matched naïve control mice (Δ); two tailed t-test. PWL=Paw Withdrawal Latency. PWR=Paw Withdrawal Response **c** Representative blots and densitometric quantification of POMC and MOR protein levels in the DRG of male mice (control, n=5 and diabetic, n=5). Data represents mean \pm SEM. * p <0.05, ** p <0.01, *** p <0.001; two tailed t-test.

Reviewer Figure 3: Pain phenotype of female and male diabetic mice at their peak sensitivities. **a, b** Mechanical and **c, d** thermal hypersensitivity of **a** female (control, n=11 and diabetic, n=12) and **b** male (control, n=8 and diabetic, n=8) diabetic mice at 12 weeks and 6 weeks post STZ-induction respectively, compared with their respective age and gender-matched controls. Data represents mean \pm SEM. * $p < 0.05$, ** $p < 0.01$, *** $p < 0.001$; two tailed t-test.

Reviewer #2 (Remarks to the Author): *This is a well-crafted story, reporting that sensory DRG neurons express POMC in mice and human and suggesting its antinociceptive role. POMC is expressed in multiple organs and different cell types, including neurons in the hypothalamus, cells in the pituitary gland, melanocytes in the skin. So far, very little is known about POMC expression in the peripheral nervous system. The authors do a very nice job demonstrating the expression of POMC in DRG in both mice and human. Moreover, the authors find that POMC-MOR axis' expression is downregulated in both the diabetic animal model and patients. This finding is very novel and interesting. Furthermore, the authors find that reconstitution of POMC in DRG rescues both thermal and mechanical hyperalgesia, suggesting its importance and clinic significance of POMC in Painful neuropathy. Although it is known MOR exists in DRG, its local endogenous ligands are not clear. Thus, this manuscript makes a substantial contribution to literature, as well as extending our understanding of PNS POMC in the control of pain sensitivity.*

Author response: We thank the reviewer for the kind words of appreciation

Major concerns:

Reviewer 2, Comment 1: *The major weakness is that given that POMC-MOR expression is downregulated in both female and male animals, why female diabetic mice are hyperalgesia while and male diabetic mice are hypoalgesia. Is there any difference between male and female in DRG POMC expression? If overexpression POMC in DRG in normal or diabetic male mice, will this also alter the pain sensitivity?*

Author response: We agree with the issue raised by the reviewer. In the revised manuscript we have performed series of new experiments addressing each question.

In our initial manuscript, to determine the time point of maximum hypersensitivity, the male mice were tested for thermal hyperalgesia using Hotplate, which is a relatively broad measure. In our revised manuscript, we performed a thorough pain phenotyping over the course of time.

We examined the male mice for mechanical hyperalgesia (using Von Frey filaments) and thermal hyperalgesia (using Hargreaves) every 2 weeks post STZ-induction and compared them with age-matched healthy controls. In our setting using a low-dose STZ protocol, a brief

thermal hypersensitivity phase was detectable using Hargreaves at 6 weeks, after which the male diabetic mice tended towards hypoalgesia (**Reviewer Figure 2a, Revised Manuscript Figure S5**). This small but significant difference in thermal hypersensitivity was detectable using Hargreaves, but not in the Hotplate method. The male mice displayed a comparatively prolonged period of mechanical hypersensitivity with a peak at 6 weeks post-induction (**Reviewer Figure 2b, Revised Manuscript Figure S7a**). The pain phenotyping we conducted for our initial manuscript was at 4, 8 and 12 weeks post-induction. We therefore had missed this brief phase of hyperalgesia at 6 weeks detectable using Hargreaves and von Frey filaments.

Having identified hypersensitivity peak at 6 weeks post-induction, we harvested the DRG and quantified POMC and MOR protein levels in the diabetic male mice. Consistent with our findings in the female diabetic mice, the POMC and MOR protein levels were significantly decreased in the DRG of male diabetic mice at this time point (**Reviewer Figure 2c, Revised Manuscript Figure S7**).

In terms of pain sensitivity, the disease progressed differently in both sexes over the course of time, with females tending towards hyperalgesia at 12 week post-induction, whilst the males displayed a hyperalgesic phase comparatively earlier. Though both genders displayed a neuropathic pain phenotype, we noted that the hypersensitivity (thermal and mechanical) was more pronounced in the female diabetic mice than in the male diabetic mice (**Reviewer Figure 3, Revised Manuscript Figure 2a, 2b, S7a, S7b**). Similar gender differences showing higher propensity of female sex to neuropathic pain and earlier onset of neuropathy in the males, have been previously reported in patients²¹⁻²⁵ and animal models of diabetic neuropathic pain^{26,27}. To investigate why there is a gender-specific difference with respect to the DPN onset and progression is beyond the scope of this study. However, the key finding is that when both male and female mice display the hypersensitivity, there is a concurrent loss of POMC and MOR regardless of the gender. This would suggest that gender difference is not responsible for this downregulation. This is further supported by the observation that the decrease of POMC or MOR was observed in both, female and male patients. This issue has been addressed on **pg 20 of the Revised Manuscript**.

Finally, we overexpressed the POMC-MOR bicistronic construct in the DRG of diabetic male mice using adeno-associated viral particles. We performed functional analyses for thermal and mechanical hypersensitivities, as well as, gait parameters at 6 weeks post STZ-induction. We observed a significant amelioration of the evoked pain responses and associated gait anomalies in the mice overexpressing POMC and MOR protein, upon comparison with the diabetic male mice expression only GFP viral constructs (**Reviewer Figure 4, Revised Manuscript Figure S11, S12 and S13**).

Thus, taken together, despite time-related differences in the genders with respect to progression of painful DPN, neuropathic pain phenotype was observed in females and males, which could be rescued upon re-instating the dysfunctional POMC-MOR axis.

Reviewer Figure 4: Genetic reconstitution of POMC and MOR alleviates neuropathic phenotype in male diabetic mice. Male diabetic mice (labelled as ‚D‘ in the Figure legend) injected with rAAV-9/2 constructs for overexpressing GFP or POMC-MOR. Readings taken after intraplantar injection of PKC inhibitor in both hindpaws (Gö6983, 20 μ M, 45 minutes). Measurements were performed 6 weeks post STZ-induction. **a** Immunostaining of AAV-injected DRG with GFP or FLAG specific antibodies showing expression of the viral constructs. **b** Mechanical (Frey filaments) **c** thermal hypersensitivity (Hargreaves) and **d** gait parameters (Catwalk) measured in both hindpaws using von in naïve control (non-diabetic, non-AAV injected), and virus injected diabetic mice. (Naïve control, n=10; D+GFP, D+POMC-MOR, n=8). For panel **b** PWR= Paw Withdrawal responses. * indicates significant difference between D+POMC-MOR mice and D+GFP mice readings. For panel **c** PWL=Paw Withdrawal Latency. Figure legend applies for panels **b**, **c** and **d**. For all panels, data represents mean \pm SEM; two-way ANOVA followed by Dunnett’s post-hoc test. Scale=50 μ m

Reviewer 2, Comment 2: Since POMC expression is downregulated under diabetic conditions, it is also interesting to know whether POMC expression will be increased in response to acute pain stimulus. That is whether DRG POMC plays an antinociceptive role under normal conditions.

Author response: To address this question, we employed the capsaicin-induced acute pain model in healthy mice (**Reviewer Figure 5, Revised Manuscript Figure S4**). We injected capsaicin in one hindpaw (ipsilateral) and vehicle in the other (contralateral). We then quantified *Pomc* mRNA and protein levels in the DRG (site of POMC synthesis) and footpads (site of POMC proteolysis and release). We observed that capsaicin evoked acute nocifensive behavior, which lasted for ca. 5 minutes post injection (**Reviewer Figure 5b**), suggesting the resolution of the acute pain thereafter. At 30 minutes post injection, we observed significantly decreased POMC protein levels in the ipsilateral footpads and DRG as

compared to contralateral tissues (**Reviewer Figure 5d**). The gene expression, however, was unchanged at this time point (**Reviewer Figure 5c**). At 4 hours post injection, the *Pomc* mRNA in the ipsilateral DRG was significantly increased compared to the contralateral DRG. This was reflected in the POMC protein level, which was normalized (**Reviewer Figure 5e**). Taken together, these findings indicate that capsaicin induces an initial release of POMC peptides from the nerves around the time the acute pain is resolved, which is then replenished by an increased *Pomc* gene expression.

This finding is supported by previous studies, which have conducted baseline pain measurements in MOR knockout mice. For instance, Weibel *et al* demonstrated increased thermal and mechanical sensitivity in MOR knockout mice²⁸. Similar observations have been made by Contet *et al*²⁹. Sora *et al* showed increased basal hypersensitivity in MOR knockout mice using tail flick assay and hotplate assay³⁰. Jointly, these findings suggest the functional link of the POMC-pathway with antinociceptive signaling in the PNS under healthy conditions.

As this experiment provided important functional evidence for the role of POMC in antinociception in the PNS, the results have been included into the revised manuscript **pg 6 of the Revised Manuscript**. We are grateful to the reviewer for this suggestion.

Reviewer Figure 5: POMC in PNS participates to resolve capsaicin-induced acute pain. **a** Schematic showing injection of capsaicin (0.3%) in one foot (ipsilateral) and vehicle in the other foot (contralateral). **b** The mice spent significant time showing nocifensive behaviour only for the capsaicin-injected ipsilateral foot during the 10 minutes observation period, after which no such behaviour is observed. **c** *Pomc* gene expression quantified in the lumbar DRG shows a significant increase 4 hours post capsaicin injection. **d** A significant loss of POMC protein is observed in the lumbar DRG and footpads 30 minutes post injection indicating release of the neuropeptide. **e** The POMC protein levels are restored at 4 hours post injection in the lumbar DRG and the footpads. For all panels male, n=6 and females, n=6; *p<0.05 **p<0.01, ***p<0.001; two tailed t-test.

Minor points:

Reviewer 2, Comment 3: Fig.1b, POMC immunostaining is weak and almost covered by the red channel, which make it difficult to see POMC expression clearly. Fig.3c, in the control panel, it seems that nearly all Tuj-1 neurons express MOR in this picture, however the quantitative data shows only 60% neurons are MOR positive. The contrast setting of images may result in statistical errors. Can the authors provide raw confocal image files? Similar concern in Fig. 2d and Fig.s2.

Author response: We agree with the reviewer's concerns and have replaced the mentioned figures with more representative images. The raw confocal image files are also provided. Please note that Fig. S2 of the initial manuscript is Fig. S1 in the revised manuscript.

Reviewer 2, Comment 4: Regarding to the specificity of POMC immunostaining in DRG, can the authors provide a positive/negative control of the antibody they used? For example, using the hypothalamic arcuate nuclei or the pituitary gland as a positive control and other brain areas as a negative control.

Author response: We are grateful to the reviewer for providing this suggestion and an opportunity to strengthen our study.

As recommended, we have performed the staining of brain sections, which shows that the anti-POMC antibodies detect signal in the neuronal somata of hypothalamus but not in the cortex region, thereby demonstrating their specificity (**Reviewer Figure 6 and Revised Manuscript Figure S3a**).

Reviewer Figure 6: Specificity of the anti-POMC antibodies shown in immunofluorescence using mouse brain Naïve control mouse brain sections immunostained using either anti-POMC a Goat (Gt; #PA518368) antibody or Rabbit (Rb; #23499) antibody detect specific signal in hypothalamus region known to express POMC, whereas no signal in neuronal somata of the hippocampal region. Inset shows a magnified image of neuronal soma.

In addition, we were also able to obtain frozen DRG tissues from *Pomc* conditional knockout (cKO) mice lacking *Pomc* to further test the antibody specificity. This mouse carries a neuronal specific deletion of *Pomc* gene, generated using cre-lox system (Snap 25 cre x *Pomc* fl/fl). Upon immunostaining the DRG of the *Pomc* cKO mice and wild-type mice, we observed a loss of POMC immunoreactivity detectable in the wild-type DRG, confirming the specificity of the anti-POMC antibodies used in this study (**Reviewer Figure 7a and Revised**

Manuscript Figure S3c). In addition, we also used total protein extracts from these tissues to verify the POMC antibody specificity in immunoblotting. The 26KDa band detectable in the total DRG lysates from the wild-type mice, was significantly diminished in the conditional knockout mice, not only validating the evidence for antibody specificity, but also corroborating that neurons are the major cell type to express POMC in the DRG under basal condition **Reviewer Figure 7b and Revised Manuscript Figure S3d).**

Reviewer Figure 7: Specificities of anti-POMC antibodies shown using a *Pomc* conditional knockout mouse DRG Specificity of the anti-POMC antibodies shown in **a** immunofluorescence and **b** immunoblot using *Pomc* conditional knockout (cKO; lacking *Pomc* expression in all neurons expressing Snap25 protein generated using cre-lox system) as negative control. **a** Naïve wild-type (wt) or cKO DRG sections immunostained using either anti-POMC a Goat (Gt; #PA518368) antibody or Rabbit (Rb; #23499) antibody detect a in wt, but not cKO DRG. **b** Representative blot and actin-normalized densitometric quantification shows the detection of ~26 KDa POMC band in the Wt DRG lysates and the > 95% loss of the same band in cKO DRG lysates, confirming the specificity of anti-POMC Gt antibody. Scale=50 μ m.

Reviewer #3 (Remarks to the Author): *In this work, Deshpande and colleagues have identified a pain-promoting pathway which involves a reduction of proopiomelanocortin (POMC) signalling (at both the ligand and receptor levels) in hyperglycaemic conditions encountered in diabetic neuropathy. The authors trace back these changes to the periphery, namely the dorsal root ganglion (DRG) and associated axons innervating the skin. Rescuing the diabetes-induced downregulation of POMC signalling via viral overexpression of the ligand and/or its receptor in the DRG provided analgesic effects across a variety of pain modalities. The authors propose this mechanism could form the basis for new antinociceptive therapies in humans.*

The study contains a significant amount of data, including extensive behavioural pain testing. The existing experiments are conducted carefully using all the necessary controls and the results are explained in detail. The addition of human samples further substantiates the data obtained in mice. The methods are written in sufficient detail to allow replication of the experiments. The statistical analysis employed for each comparison is denoted clearly and is appropriate. Overall, the paper is very well written – experiments follow a logical flow and the narrative is clear. The conclusions are supported by the experimental data and the findings are novel.

While the results are interesting and solid, there is some key information missing which could extend the applicability of the conclusions and make it more translational.

Author response: We thank the reviewer for the kind words of appreciation and for providing suggestions that have helped expand the scope of the study.

Reviewer 3, Comment 1: *The authors focused on female diabetic mice on the basis that this sex develops thermal hypersensitivity (as opposed to hyposensitivity in males) at the examined time points after diabetes induction. Given the recently emerged knowledge on sexual dimorphism in pain responses, it is important to characterise this pathway a bit more with regards to sex. Obvious questions are:*

- Does the downregulation of POMC and MOR occur in the same way in male diabetic mice?
- Does viral overexpression of POMC/MOR reverse pain behaviours in male diabetic mice?
This may not be possible to assess for thermal hypersensitivity, but it is definitely an option for mechanical pain as well as spontaneous pain tests.

Author response: We agree with the reviewer and have conducted a series of new experiments to answer all the questions asked.

In our initial manuscript, to determine the time point of maximum hypersensitivity, the male mice were tested for thermal hyperalgesia using Hotplate, which is a relatively broad measure. In our revised manuscript, we performed a thorough pain phenotyping over the course of time.

We examined the male mice for mechanical hyperalgesia (using Von Frey filaments) and thermal hyperalgesia (using Hargreaves) every 2 weeks post STZ-induction and compared them with age-matched healthy controls. In our setting using a low-dose STZ protocol, a brief thermal hypersensitivity phase was detectable using Hargreaves at 6 weeks, after which the male diabetic mice tended towards hypoalgesia (**Reviewer Figure 2a, Revised Manuscript Figure S5**). This small but significant difference in thermal hypersensitivity was detectable using Hargreaves, but not in the Hotplate method. The male mice displayed a comparatively prolonged period of mechanical hypersensitivity with a peak at 6 weeks post-induction (**Reviewer Figure 2b, Revised Manuscript Figure S7a**). The pain phenotyping we conducted for our initial manuscript was at 4, 8 and 12 weeks post-induction. We therefore had missed this brief phase of hyperalgesia at 6 weeks detectable using Hargreaves and von Frey filaments.

Having identified hypersensitivity peak at 6 weeks post-induction, we harvested the DRG and quantified POMC and MOR protein levels in the diabetic male mice. Consistent with our findings in the female diabetic mice, the POMC and MOR protein levels were significantly decreased in the DRG of male diabetic mice at this time point (**Reviewer Figure 2c, Revised Manuscript Figure S7**).

In terms of pain sensitivity, the disease progressed differently in both sexes over the course of time, with females tending towards hyperalgesia at 12 week post-induction, whilst the males displayed a hyperalgesic phase comparatively earlier. Though both genders displayed a neuropathic pain phenotype, we noted that the hypersensitivity (thermal and mechanical) was more pronounced in the female diabetic mice than in the male diabetic mice (**Reviewer Figure 3, Revised Manuscript Figure 2a, 2b, S7a, S7b**). Similar gender differences showing higher propensity of female sex to neuropathic pain and earlier onset of neuropathy in the males, have been previously reported in patients²¹⁻²⁵ and animal models of diabetic

neuropathic pain^{26,27}. To investigate why there is a gender-specific difference with respect to the DPN onset and progression is beyond the scope of this study. However, the key finding is that when both male and female mice display the hypersensitivity, there is a concurrent loss of POMC and MOR regardless of the gender. This would suggest that gender difference is not responsible for this downregulation. This is further supported by the observation that the decrease of POMC or MOR was observed in both, female and male patients. has been addressed on **pg 20 of the Revised Manuscript**.

Finally, we overexpressed the POMC-MOR bicistronic construct in the DRG of diabetic male mice using adeno-associated viral particles. We performed functional analyses for thermal and mechanical hypersensitivities, as well as, gait parameters at 6 weeks post STZ-induction. We observed a significant amelioration of the evoked pain responses and associated gait anomalies in the mice overexpressing POMC and MOR protein, upon comparison with the diabetic male mice expression only GFP viral constructs (**Reviewer Figure 4, Revised Manuscript Figure S11, S12 and S13**).

Thus, taken together, despite small differences in the genders with respect to progression of painful DPN, neuropathic pain phenotype was observed in females and males, which could be rescued upon re-instating the dysfunctional POMC-MOR axis.

Reviewer 3, Comment 2: *The authors conclude that the site of interest is the periphery where they detect downregulation of the ligand/receptor. The evidence for this seems clear. However we cannot exclude an additional site of action in the CNS, where MOR is also expressed.*

-Is there evidence for downregulation of this pathway in the spinal cord?

-Viral overexpression in DRG has antinociceptive effects but can the authors exclude that a) ligand is transported to central terminals to act on postsynaptic MOR in the spinal cord b) the injected AAV does not transduce spinal cord neurons?

Author response: We thank the reviewer for raising a valid point.

Exogenous MOR agonists (e.g. morphine derivatives) are widely used to treat acute and chronic pain. However, previous studies have shown that painful DPN is often resistant to opioid analgesics^{31,32}. To explore the underlying reason, previous studies have determined the levels and/or functionality of MOR in the spinal cord and brain. Chen *et al*, Zurek *et al* and Shakura *et al* observed a decreased responsiveness to intrathecally injected MOR agonists, but the antinociceptive effect of the same agonists given intracerebrally was maintained in these diabetic animal models³²⁻³⁵. This was associated with a functional downregulation of MOR in the spinal cord during diabetes. Shakura *et al* also observed decreased MOR protein levels in the lumbar spinal cord in diabetic rats³⁵. However, other studies have shown that the decreased level and/or functionality of spinal MOR during neuropathic pain conditions is possibly associated with the administration of exogenous MOR agonist itself^{36,37}, which causes MOR desensitization, degradation and induces opioid tolerance³⁸.

In our study, we demonstrated that the MOR protein level was decreased in the PNS of diabetic mice in the absence of any exogenous application of MOR agonists. This finding, along with the detection of the MOR ligand derived from POMC in the PNS was interesting, since this suggested that neuropathic pain could be possibly managed at the PNS level,

without any adverse side-effects (such as tolerance and addiction) involving the CNS. Therefore, our main focus was the peripheral sensory neurons and the modulation of PNS intrinsic pathway to rescue painful DPN.

For this purpose, we have mainly used direct DRG injections in which, the AAV constructs were injected in L3 and L4 DRG only. Such a delivery technique allows viral expression in all neurons of the DRG and their axons projecting in the sciatic nerve, as well as, those in those going to the spinal cord. Previous studies have shown that virus injected using this technique is not transduced to the spinal neurons³⁹⁻⁴¹. We have also used intrathecal delivery of the AAV particles by injecting at the level of L4-L5 region of the spinal cord (Revised manuscript Fig S10, S11 and S12). This delivery method has been shown to express the virus maximally in the DRG neurons, minimally in the spinal cord and negligibly in the brain⁴².

Nevertheless, given the expression of MOR in spinal cord and brain, and of POMC in hypothalamus, we agree that further exploration of this pathway in the CNS is interesting, although that would constitute a new study with a new focus.

Reviewer 3, Comment 3: *It would be very interesting to ascertain whether this pathway is specific to metabolic neuropathies such as diabetic neuropathy, or a more generalised mechanism that also extends to traumatic neuropathies. The authors could use a model such as spinal nerve ligation to quickly check whether POMC/MOR downregulation also occurs in the context.*

Author response: In order to further explore the POMC-MOR mechanism in other neuropathies, we performed molecular analysis of the DRG of mice with spared nerve injury (SNI)¹⁴. We measured POMC and MOR protein levels in the L3 and L4 DRG of the ipsilateral and contralateral side of SNI and sham-operated mice at 7 days post-operation. No changes were observed in the DRG of the sham-operated mice. Surprisingly, POMC and MOR were both downregulated in the ipsilateral DRG of SNI mice (**Reviewer Figure 1**). The downregulation of MOR in the DRG after SNI has been previously reported by Wieskopf *et al*¹⁵. However, our preliminary findings show that the malfunction exists also upstream of the receptor, *i.e.* at the ligand level. To investigate the mechanisms underlying as the downregulation of POMC-MOR in SNI, is beyond the scope of this study. The well-studied chronic neuroinflammatory changes (including NF- κ B¹⁶⁻¹⁸ and PKC^{19,20} activation) occurring in the DRG during SNI could be a contributing factor. These findings present an interesting basis for future studies with different focus.

Taken together, our preliminary findings in the SNI mouse model show that the POMC-pathway may represent a more generalized mechanism and possibly be extended to other chronic neuroinflammatory conditions.

Reviewer 3, Comment 4: *While the PKC-mediated degradation of MOR is a plausible mechanism, there could be other pathways at play eg phosphorylation of ion channels. The authors should acknowledge this possibility in the discussion. Do PKA inhibitors also have an effect on MOR internalisation and/or diabetic pain?*

Author response: The elevation of PKC has indeed been linked to many other pathways causing neuropathic pain, as rightly pointed out by the reviewer. Previous studies by Struder *et al* and Vellani *et al* have shown that phosphorylation of TRPV1^{43,44} by PKC in the PNS leads to increase in ion channel activity linked with neuropathic pain. Similar observations have also been for the Nav1.8⁶ ion channel, purinergic receptor (P2X3)⁴⁵. We have included

a discussion on the broad implications of PKC activation on **pg 19 of the Revised Manuscript**.

PKA, like PKC, is a cAMP-dependant kinase and is also activated during diabetes¹¹. That PKA can phosphorylate ion channels leading to neuronal hyperexcitability has been shown^{43,46,47}. PKA inhibitors have also been shown to improve painful DPN⁴⁸.

Within the context of the POMC signalling loop, PKA-mediated MOR phosphorylation occurs when MOR binds to its agonist peptide (homologous phosphorylation). During diabetes, we detected lowered β -endorphin level, suggesting that heterologous phosphorylation is a more likely event. We also observed an increased phosphorylation at the threonine residues of MOR protein in the DRG of diabetic mice. Illing *et al* have shown that PKA does not participate in heterologous phosphorylation of threonine residues⁴⁹.

As such, although contribution of PKA to neuropathic phenotype is possible, based on our observations, PKC is more relevant with respect to the POMC-MOR axis during diabetes.

Some more minor points:

Reviewer 3, Comment 5: *While the authors provide a control staining (pre-incubation with antigen) to demonstrate the specificity of the POMC antibody, unfortunately this control is of limited use as it only shows that the antibody correctly recognises this antigen – it is however still possible that it also recognises distinct but similar proteins. The ideal control would be testing the antibody on tissue taken from POMC KO mice, which appear to be available (e.g. <https://doi.org/10.1073/pnas.0306931101> and <https://www.jax.org/strain/003191>). If the authors get some of these mice, it would also be very interesting to check their baseline pain responses as well as how they develop diabetic pain.*

Author response: We concur with the reviewer on this issue. We have now included better controls showing anti-POMC antibody specificity. As suggested by reviewer 2, we have immunostained the hypothalamic region of the brain known to express POMC as a positive control and the cortex region as negative control. Using the anti-POMC antibodies, we observed a clear signal in the neuronal somata of the hypothalamic region, but not the cortex region, thereby showing the specificity of the anti-POMC antibodies (**Reviewer Figure 6 and Revised Manuscript Figure S3a**).

Since the *Pomc* KO mice suggested by the reviewer were available only as frozen embryos (Jackson), we were unable to obtain and resurrect them within the time frame of this revision period. Due to unavailability of the live *Pomc* KO or the cKO mice, we could not perform the pain measurements in these mice.

However, we were able to obtain frozen DRG tissues from *Pomc* conditional knockout (cKO) mice lacking POMC to test the antibody specificity. This mouse carries a neuron-specific deletion of *Pomc* gene, generated using cre-lox system (Snap 25 cre x *Pomc* fl/fl). Upon immunostaining the DRG of the *Pomc* cKO mice and wild-type mice, we observed a loss of POMC immunoreactivity detectable in the wt DRG, confirming the specificity of the anti-POMC antibodies used in this study (**Reviewer Figure 7a and Revised Manuscript Figure S3c**). In addition, we also used these the tissues from these mice to verify POMC antibody specificity in immunoblotting. The 26KDa band detectable in the total DRG lysates of wild-

type mice, was significantly diminished in the conditional knockout mice, not only validating the evidence for antibody specificity, but also corroborating that neurons are the major cell type to express POMC in the DRG under basal condition **Reviewer Figure 7b and Revised Manuscript Figure S3d**).

We are grateful to reviewer for providing this suggestion and an opportunity to strengthen our study.

Reviewer 3, Comment 6: *The presented work identifies a pain-protective pathway that could be exploited for treatment of diabetic pain, but presumably the POMC/MOR downregulation is not the source of diabetic pain. It would be useful to have a very brief discussion of established and emerging candidate molecules that could be responsible for the pathogenesis of pain such as Nav1.8, HCN2, TRPV1, Cav, especially since there might be overlap of pathways involving cAMP/PKA/PKC*

(eg <https://stm.sciencemag.org/content/9/409/eaam6072>, <https://www.nature.com/articles/nm.2750>, -,mnbvc)

Reviewer 3, Comment 7: *In terms of applicability of results in the clinic, the authors could perhaps extend the discussion. How are these results likely to change the already established use of opioids for pain control in diabetes? Since we need to restore expression of ligand/receptor, would the proposed way forward be introducing copies of the protein via gene therapy?*

Author response: We thank the reviewer for the above two suggestions, both of which have helped to emphasize the translational nature of the findings.

This discussion suggested by the reviewer is **given below and on pg 21 of the Revised manuscript**.

Studies reporting novel disease mechanisms are currently driving the momentum of developing pathogenesis-oriented therapeutics for painful DPN. *E.g.* blocking the HCN2 channels using ivabradine⁴⁸ or scavenging methylglyoxal and preventing Na_v1.8 glycation⁴ have shown promise in *in vivo* studies. Other emerging molecules include the selective sodium and calcium channel blockers, TRPA1 antagonists among others⁵⁰. While most of these approaches target the excitatory ion channels/receptors, we propose a novel strategy of restoring body's natural defense mechanism of controlling pain signaling. It provides a platform for the development of new therapies by boosting the endogenous POMC synthesis and prevention of MOR degradation. Administration of *Pomc* promoter agonists, such as CRH (routinely administered in the patients with Cushing's disease of POMC deficiency)⁵¹ or melanocortin receptor agonists into the spinal cord may be one of the possibilities to enhance the β -endorphin level in the PNS. Other compounds, such as the melanocortin receptor agonists, which have shown to enhance POMC expression in patients with genetic POMC deficiency^{52,53}, may be another promising option. Gene delivery techniques would also be a potential therapeutic in the future for patients with painful DPN. Lastly, in light of the ongoing opioid crisis, our study offers a promising alternative approach to counter peripheral neuropathic pain.

References

1. Suzuki, Y., Sato, J., Kawanishi, M. & Mizumura, K. Lowered response threshold and increased responsiveness to mechanical stimulation of cutaneous nociceptive fibers in streptozotocin-diabetic rat skin in vitro—correlates of mechanical allodynia and hyperalgesia observed in the early stage of diabetes. *Neurosci. Res.* **43**, 171–178 (2002).
2. Ørstavik, K. *et al.* Abnormal Function of C-Fibers in Patients with Diabetic Neuropathy. *J. Neurosci.* **26**, 11287–11294 (2006).
3. Blair, N. T. & Bean, B. P. Roles of Tetrodotoxin (TTX)-Sensitive Na⁺ Current, TTX-Resistant Na⁺ Current, and Ca²⁺ Current in the Action Potentials of Nociceptive Sensory Neurons. *J. Neurosci.* **22**, 10277–10290 (2002).
4. Bierhaus, A. *et al.* Methylglyoxal modification of Na^v 1.8 facilitates nociceptive neuron firing and causes hyperalgesia in diabetic neuropathy. *Nat. Med.* **18**, 926–933 (2012).
5. Orestes, P. *et al.* Reversal of Neuropathic Pain in Diabetes by Targeting Glycosylation of Cav3.2 T-Type Calcium Channels. *Diabetes* **62**, 3828–3838 (2013).
6. Wu, D.-F. *et al.* PKC ϵ phosphorylation of the sodium channel NaV1.8 increases channel function and produces mechanical hyperalgesia in mice. *J. Clin. Invest.* **122**, 1306–1315 (2012).
7. Zenker, J. *et al.* Altered Distribution of Juxtaparanodal Kv1.2 Subunits Mediates Peripheral Nerve Hyperexcitability in Type 2 Diabetes Mellitus. *J. Neurosci.* **32**, 7493–7498 (2012).
8. Woolf, C. J. Central sensitization: Implications for the diagnosis and treatment of pain. *Pain* **152**, S2-15 (2011).
9. Marshall, A. G. *et al.* Spinal Disinhibition in Experimental and Clinical Painful Diabetic Neuropathy. *Diabetes* **66**, 1380–1390 (2017).
10. Tesfaye, S., Boulton, A. J. M. & Dickenson, A. H. Mechanisms and Management of Diabetic Painful Distal Symmetrical Polyneuropathy. *Diabetes Care* **36**, 2456–2465 (2013).
11. Vincent, A. M., Callaghan, B. C., Smith, A. L. & Feldman, E. L. Diabetic neuropathy: cellular mechanisms as therapeutic targets. *Nat. Rev. Neurol.* **7**, 573–583 (2011).
12. Frank, T., Nawroth, P. & Kuner, R. Structure-function relationships in peripheral nerve contributions to diabetic peripheral neuropathy. *Pain* **160 Suppl 1**, S29–S36 (2019).
13. Schwarz, D. *et al.* Characterization of experimental diabetic neuropathy using multicontrast magnetic resonance neurography at ultra high field strength. *Sci. Rep.* **10**, 1–12 (2020).
14. Decosterd, I. & Woolf, C. J. Spared nerve injury: an animal model of persistent peripheral neuropathic pain. *PAIN* **87**, 149–158 (2000).
15. Wieskopf, J. S. *et al.* Broad spectrum analgesic efficacy of IBNtxA is mediated by exon 11-associated splice variants of the mu-opioid receptor gene. *Pain* **155**, 2063–2070 (2014).
16. Zeng, Y. *et al.* Reduction of Silent Information Regulator 1 Activates Interleukin-33/ST2 Signaling and Contributes to Neuropathic Pain Induced by Spared Nerve Injury in Rats. *Front. Mol. Neurosci.* **13**, (2020).
17. Xia, Y., Xue, M., Wang, Y., Huang, Z. & Huang, C. Electroacupuncture Alleviates Spared Nerve Injury-Induced Neuropathic Pain And Modulates HMGB1/NF- κ B Signaling Pathway In The Spinal Cord. *J. Pain Res.* **12**, 2851–2863 (2019).
18. Ma, W. & Bisby, M. A. Increased activation of nuclear factor kappa B in rat lumbar dorsal root ganglion neurons following partial sciatic nerve injuries. *Brain Res.* **797**, 243–254 (1998).
19. Ko, M.-H., Yang, M.-L., Youn, S.-C., Lan, C.-T. & Tseng, T.-J. Intact subepidermal nerve fibers mediate mechanical hypersensitivity via the activation of protein kinase C gamma in spared nerve injury. *Mol. Pain* **12**, (2016).
20. Gu, Y., Li, G., Chen, Y. & Huang, L.-Y. M. Epac–protein kinase C alpha signaling in purinergic P2X3R-mediated hyperalgesia after inflammation. *Pain* **157**, 1541–1550 (2016).
21. Cardinez, N. *et al.* Sex differences in neuropathic pain in longstanding diabetes: Results from the Canadian Study of Longevity in Type 1 Diabetes. *J. Diabetes Complications* **32**, 660–664 (2018).
22. Belfer, I. Sex-Specific Genetic Control of Diabetic Neuropathic Pain Suggests Subsequent Development of Men-only and Women—Only Analgesic Strategies. *EBioMedicine* **2**, 1280 (2015).
23. Abbott, C. A., Malik, R. A., van Ross, E. R. E., Kulkarni, J. & Boulton, A. J. M. Prevalence and characteristics of painful diabetic neuropathy in a large community-based diabetic population in the U.K. *Diabetes Care* **34**, 2220–2224 (2011).
24. Sorensen, L., Molyneaux, L. & Yue, D. K. Insensate versus painful diabetic neuropathy: the effects of height, gender, ethnicity and glycaemic control. *Diabetes Res. Clin. Pract.* **57**, 45–51 (2002).
25. Aaberg, M. L., Burch, D. M., Hud, Z. R. & Zacharias, M. P. Gender differences in the onset of diabetic neuropathy. *J. Diabetes Complications* **22**, 83–87 (2008).
26. Pesaresi, M. *et al.* Axonal transport in a peripheral diabetic neuropathy model: sex-dimorphic features. *Biol. Sex Differ.* **9**, 6 (2018).

27. Joseph, E. K. & Levine, J. D. Sexual dimorphism in the contribution of protein kinase c isoforms to nociception in the streptozotocin diabetic rat. *Neuroscience* **120**, 907–913 (2003).
28. Weibel, R. *et al.* Mu Opioid Receptors on Primary Afferent Nav1.8 Neurons Contribute to Opiate-Induced Analgesia: Insight from Conditional Knockout Mice. *PLoS ONE* **8**, e74706 (2013).
29. Contet, C. *et al.* Dissociation of Analgesic and Hormonal Responses to Forced Swim Stress Using Opioid Receptor Knockout Mice. *Neuropsychopharmacology* **31**, 1733–1744 (2006).
30. Sora, I. *et al.* Opiate receptor knockout mice define μ receptor roles in endogenous nociceptive responses and morphine-induced analgesia. *Proc. Natl. Acad. Sci. U. S. A.* **94**, 1544–1549 (1997).
31. Ziegler, D. & Fonseca, V. From guideline to patient: a review of recent recommendations for pharmacotherapy of painful diabetic neuropathy. *J. Diabetes Complications* **29**, 146–156 (2015).
32. Zurek, J. R., Nadeson, R. & Goodchild, C. S. Spinal and supraspinal components of opioid antinociception in streptozotocin induced diabetic neuropathy in rats. *Pain* **90**, 57–63 (2001).
33. Chen, S.-R. & Pan, H.-L. Antinociceptive effect of morphine, but not mu opioid receptor number, is attenuated in the spinal cord of diabetic rats. *Anesthesiology* **99**, 1409–1414 (2003).
34. Chen, S.-R., Sweigart, K. L., Lakoski, J. M. & Pan, H.-L. Functional mu opioid receptors are reduced in the spinal cord dorsal horn of diabetic rats. *Anesthesiology* **97**, 1602–1608 (2002).
35. Shaqura, M. *et al.* Reduced number, G protein coupling, and antinociceptive efficacy of spinal mu-opioid receptors in diabetic rats are reversed by nerve growth factor. *J. Pain Off. J. Am. Pain Soc.* **14**, 720–730 (2013).
36. Courteix, C. *et al.* Is the Reduced Efficacy of Morphine in Diabetic Rats Caused by Alterations of Opiate Receptors or of Morphine Pharmacokinetics? *J. Pharmacol. Exp. Ther.* **285**, 63–70 (1998).
37. Tiwari, V. *et al.* Peripherally acting mu-opioid receptor agonists attenuate ongoing pain-associated behavior and spontaneous neuronal activity after nerve injury in rats. *Anesthesiology* **128**, 1220–1236 (2018).
38. Corder, G. *et al.* Loss of μ opioid receptor signaling in nociceptors, but not microglia, abrogates morphine tolerance without disrupting analgesia. *Nat. Med.* **23**, 164–173 (2017).
39. Storek, B. *et al.* Sensory neuron targeting by self-complementary AAV8 via lumbar puncture for chronic pain. *Proc. Natl. Acad. Sci. U. S. A.* **105**, 1055–1060 (2008).
40. Xu, Y., Gu, Y., Wu, P., Li, G.-W. & Huang, L.-Y. M. Efficiencies of Transgene Expression in Nociceptive Neurons Through Different Routes of Delivery of Adeno-Associated Viral Vectors. *Hum. Gene Ther.* **14**, 897–906 (2003).
41. Glatzel, M. *et al.* Adenoviral and adeno-associated viral transfer of genes to the peripheral nervous system. *Proc. Natl. Acad. Sci. U. S. A.* **97**, 442–447 (2000).
42. Storek, B. *et al.* Intrathecal long-term gene expression by self-complementary adeno-associated virus type 1 suitable for chronic pain studies in rats. *Mol. Pain* **2**, 4 (2006).
43. Studer, M. & McNaughton, P. A. Modulation of single-channel properties of TRPV1 by phosphorylation. *J. Physiol.* **588**, 3743–3756 (2010).
44. Vellani, V. *et al.* Protease activated receptors 1 and 4 sensitize TRPV1 in nociceptive neurones. *Mol. Pain* **6**, 61 (2010).
45. Zhou, Y. *et al.* Suppressing PKC-dependent membrane P2X3 receptor upregulation in dorsal root ganglia mediated electroacupuncture analgesia in rat painful diabetic neuropathy. *Purinergic Signal.* **14**, 359–369 (2018).
46. Nuwer, M. O., Picchione, K. E. & Bhattacharjee, A. PKA-Induced Internalization of Slack KNa Channels Produces Dorsal Root Ganglion Neuron Hyperexcitability. *J. Neurosci.* **30**, 14165–14172 (2010).
47. Breiting, U. *et al.* PKA and PKC Modulators Affect Ion Channel Function and Internalization of Recombinant Alpha1 and Alpha1-Beta Glycine Receptors. *Front. Mol. Neurosci.* **11**, (2018).
48. Tsantoulas, C. *et al.* Hyperpolarization-activated cyclic nucleotide-gated 2 (HCN2) ion channels drive pain in mouse models of diabetic neuropathy. *Sci. Transl. Med.* **9**, eaam6072 (2017).
49. Illing, S., Mann, A. & Schulz, S. Heterologous regulation of agonist-independent μ -opioid receptor phosphorylation by protein kinase C. *Br. J. Pharmacol.* **171**, 1330–1340 (2014).
50. Papanas, N. & Ziegler, D. Emerging drugs for diabetic peripheral neuropathy and neuropathic pain. *Expert Opin. Emerg. Drugs* **21**, 393–407 (2016).
51. Colao, A. *et al.* Corticotropin-releasing hormone administration increases alpha-melanocyte-stimulating hormone levels in the inferior petrosal sinuses in a subset of patients with Cushing's disease. *Horm. Res.* **46**, 26–32 (1996).
52. Kühnen, P. *et al.* Proopiomelanocortin Deficiency Treated with a Melanocortin-4 Receptor Agonist. *N. Engl. J. Med.* **375**, 240–246 (2016).
53. Müller, T. D., Tschöp, M. H. & O'Rahilly, S. Metabolic Precision Medicines: Curing POMC Deficiency. *Cell Metab.* **24**, 194–195 (2016).

REVIEWERS' COMMENTS

Reviewer #1 (Remarks to the Author):

My queries have been answered. The manuscript has substantially improved.

Reviewer #2 (Remarks to the Author):

The authors have addressed all my concerns with new experiments in the revised manuscript. I support the publication of this manuscript.

Reviewer #3 (Remarks to the Author):

All my concerns/suggestions have been thoroughly addressed and I am happy to recommend publication of this important study.

Christoforos Tsantoulas